# The Recovery of Epidermal Proliferation Pattern in Human Skin Xenograft

**DOI:** 10.3390/cells14060448

**Published:** 2025-03-17

**Authors:** Olga Cherkashina, Alexandra Tsitrina, Danila Abolin, Elena Morgun, Anastasiya Kosykh, Marat Sabirov, Ekaterina Vorotelyak, Ekaterina Kalabusheva

**Affiliations:** 1Koltzov Institute of Developmental Biology, Russian Academy of Sciences, 119334 Moscow, Russiakalabusheva.e@gmail.com (E.K.); 2Ilse Katz Institute of Nanoscale Science, Ben Gurion University of the Negev, Beer Sheva 8410501, Israel; 3Research Institute of Molecular and Cellular Medicine, Peoples’ Friendship University of Russia (RUDN University), 117198 Moscow, Russia; 4Center for Precision Genome Editing and Genetic Technologies for Biomedicine, Pirogov Russian National Research Medical University, 117997 Moscow, Russia

**Keywords:** keratinocyte proliferation, skin regeneration, skin stem cells, human skin xenograft

## Abstract

Abnormalities in epidermal keratinocyte proliferation are a characteristic feature of a range of dermatological conditions. These include hyperproliferative states in psoriasis and dermatitis as well as hypoproliferative states in chronic wounds. This emphasises the importance of investigating the proliferation kinetics under conditions of healthy skin and identifying the key regulators of epidermal homeostasis, maintenance, and recovery following wound healing. Animal models contribute to our understanding of human epidermal self-renewal. Human skin xenografting overcomes the ethical limitations of studying human skin during regeneration. The application of this approach has allowed for the identification of a single population of stem cells and both slowly and rapidly cycling progenitors within the epidermal basal layer and the mapping of their location in relation to rete ridges and hair follicles. Furthermore, we have traced the dynamics of the proliferation pattern reorganization that occurs during epidermal regeneration, underlining the role of YAP activity in epidermal relief formation.

## 1. Introduction

The maintenance of skin epidermal homeostasis is dependent on the equilibrium between the proliferation of germinative keratinocytes in the basal layer and their differentiation in the suprabasal layers. This is followed by migration into the cornified layers and subsequent desquamation. Nevertheless, the specific dynamics of epidermal renewal have been a subject of contention for decades. The classical model proposed by Potten [1] suggests that the skin epidermis is constructed from epidermal proliferation units, which are column-shaped structures that contain a quiescent epidermal stem cell at a base that asymmetrically divides and produces transit-amplifying cells that undergo a series of divisions, thereby providing a differentiated cell mass. The proliferation analysis was conducted using nucleotide analogues (BrdU, EdU, etc.). Transit-amplifying cells dilute this label during multiplication, whereas rarely dividing stem cells accumulate it for a long time. Consequently, the latter were identified as label-retaining cells (LRCs). Nevertheless, an increasing number of studies on the mouse epidermis have called into question the concept of the epidermal proliferation unit. Firstly, classical LRCs that were maintained in adult mouse skin for months were identified in the hair follicle (HF) bulge area but not in the interfollicular epidermis (IFE) [2]. Subsequently, the advancement of molecular and cellular biology techniques has yielded a plethora of methodologies, including genetic manipulation and high-resolution microscopy, that, when employed in conjunction with clonal analysis, facilitate the identification of dividing cells in the basal layer, the initiation of differentiation processes in their progeny, and the classification of cell divisions as symmetrical or asymmetrical. The accumulated data have been applied to mathematical modelling, resulting in the development of an alternative hypothesis of a single progenitor. This hypothesis postulates that all keratinocytes of the basal epidermal layer are able to randomly generate new proliferating or differentiating cells [3,4,5]. Other studies remain inclined towards the existence of a hierarchy during epidermal differentiation, with the segregation of stem and committed progenitor cells [6,7]. Furthermore, two types of stem cell populations with different durations of the cell cycle were identified, located in distinct skin territories associated with specific derivatives, namely scales in the mouse tail and HFs in the mouse back skin [7,8].

The structure and physiology of human and mouse skin exhibit notable differences, which complicate the extrapolation of IFE organization principles from animal models. This highlights the necessity for rigorous validation to ensure the reliability of such extrapolations. Furthermore, the investigation of human skin self-renewal is constrained by methodological limitations, with the majority of experimental findings obtained through the analysis of biopsies and cell culture approaches. The culture of keratinocytes has been shown to produce three distinct types of colonies, exhibiting varying proliferation and differentiation rates. These observations align with the existence of stem and committed progenitor cells [9,10]. The isolation of human basal keratinocytes, followed by specific marker analysis and cell culturing, has enabled the identification of a phenotype specific to epidermal stem-like cells [11,12]. The spatial segregation of human keratinocytes is dependent on their localization within rete ridges, which are epidermal undulations situated between the papillae of the human dermis. The presence of keratinocytes with stem cell properties has been identified at the tips of rete ridges [13,14]. Additionally, Tumbar’s group has recently predicted the location of slow-dividing cells with LRC characteristics in the inter-ridge territory of human epidermis by applying a marker panel from mouse tail scale and interscale regions [8]. This discrepancy in proliferation dynamics and its interconnection with rete ridges and inter-ridges in the human epidermis necessitates a comprehensive analysis.

The in vivo tracing of human epidermal cells would provide valuable insights into proliferation patterns. This is particularly relevant given ongoing debates about stem cell dynamics, such as the existence of spatially distinct stem cell populations with different proliferation rates [8] or the hypothesis of a single progenitor [15]. While vital tracers have enabled the identification of proliferating cells and informed conclusions about their dynamics, and mathematical modelling based on computational analysis of H2B-traced epidermal cells has described proliferation patterns in the epidermis [15], these studies have primarily relied on mouse models combined with human biopsy analysis. However, these models do not allow for real-time proliferation tracing in human skin, and biopsies only capture a static state. This underscores the necessity of developing in vivo tracing approaches in human tissues. In combination with computational analysis of the obtained high-resolution images, such approaches may extend the understanding of epidermal regeneration mechanisms.

The grafting of human skin into immunodeficient mice has shown the restoration of most skin structures, confirmed the comparative biomechanical and physiological properties of the xenograft, and therefore provided a functional in vivo model for the investigation of human skin [16,17,18]. The xenograft has served as a representative model for studying pathologies [19,20,21,22,23,24], treatment options [25,26,27,28], and the processes occurring under normal conditions [29]. Our research group has developed a full-thickness human skin xenograft that comprises the epidermis, all dermal layers, adipose tissue, HFs, sebaceous glands, and sweat glands, thereby achieving the complete restoration of the epidermal stem cell niche structure [30,31]. The restoration of normal morphology supports the use of the xenograft as a suitable representation of skin structure. In contrast to the mouse model, the xenograft mimics human regeneration dynamics and provides a reliable microenvironment in comparison to cellular 2D and 3D models. We selected and analysed specific nuclear features, tested their potential to characterise stemness and proliferation capacity of epidermal keratinocytes, and then selected the clustering parameters for machine learning algorithms to analyse skin sections stained for BrdU and Ki67. This was performed to examine keratinocyte heterogeneity and proliferation dynamics in the epidermal basal layer in xenografts that have reached a homeostatic state and those undergoing regeneration.

## 2. Materials and Methods

### 2.1. Human Skin Xenotransplantation

Human scalp or abdominal skin biopsies were obtained from the Herzen Moscow Research Oncological Institute following plastic surgeries, with the written informed consent of all patients. These samples consisted of all skin layers, including adipose tissue. The biopsies were rinsed in Hanks’ solution (PanEco, Moscow, Russia) with the addition of 3 mg/mL of gentamicin and cut into 5 × 15 mm strips for graft preparation. One strip from each biopsy was frozen using liquid nitrogen in O.C.T. medium (Sakura Finetek, Torrance, CA, USA) as a control sample.

All work with mice was conducted in a specific pathogen-free (SPF) environment. All the experiments were carried out in accordance with the Guide for the Care and Use of Laboratory Animals of the Ethical Committee of the Koltzov Institute of Developmental Biology of the Russian Academy of Sciences (protocol number 71, 29 June 2023), in compliance with European Directive 2010/63/EU on the protection of experimental animals. The study used 8-week-old NOD SCID mice obtained from Charles River Laboratories. The animals were weighed, anaesthetised intraperitoneally with 2,2,2-tribromoethanol (Sigma-Aldrich, St. Louis, MO, USA) (250 mg per kg of body weight), and locally anaesthetised with carprofen. A 15 mm long transverse incision was made on the dorsal side between the shoulder blades. A full-thickness human skin strip was placed in the wound area, with the lower edge of the mouse skin incision sutured to the strip., The free edge of the mouse skin was then pulled over the graft to prevent drying and securely fixed.

A total of 15 mice were used in the study, with 5 animals for each timepoint. BrdU was administered intraperitoneally at a dose of 200 mg per kg of body weight one week prior to the collection of material. Biopsies were taken on post-grafting days (PG) 40, 75, and 110. The human skin was excised with a fragment of surrounding mouse tissue. The biopsies were mounted in O.C.T. medium (Sakura Finetek, USA), frozen in liquid nitrogen, and stored at −70 °C.

### 2.2. Primary Keratinocyte Isolation and Culture

Human scalp or abdominal skin biopsies were obtained from the Herzen Moscow Research Oncological Institute after plastic surgeries after the written informed consent of all patients. Biopsy specimens were stored and delivered to the laboratory in DMEM medium (PanEco, Moscow, Russia) containing 0.2 mg/mL gentamicin (Bio Farm Garant, Vladimir, Russia). Prior to cell isolation, skin samples were washed with Hank’s solution (PanEko, Russia) with 0.4 mg/mL gentamicin. For keratinocytes isolation, the epidermal layer was separated from the skin by incubation in 0.2% dispase solution (Sigma-Aldrich, St. Louis, MO, USA) at 4 °C overnight. The epidermal sheet was disrupted by trypsinization for 15 min in PBS solution with 0.25% trypsin (Gibco, New York, NY, USA) in a 1:1 ratio. KC suspension was inoculated in DMEM/F12 medium (PanEko, Russia) containing 4 mM glutamine (Gibco, USA), 10% foetal bovine serum (FBS) (HyClone, Wilmington, DE, USA), 10 ng/mL EGF (Sigma-Aldrich, St. Louis, MO, USA), 5 mg/mL insulin (Sigma-Aldrich), and 0.25 mg/mL isoproterenol (Sigma-Aldrich).

To analyse the influence of YAP activation, keratinocytes were cultured in DMEM/F12 medium containing 4 mM glutamine (Gibco, USA) and 1% foetal bovine serum (FBS) (HyClone), supplemented with 1X Insulin-Transferrin-Selenium (Thermo Fisher Scientific, Waltham, MA, USA) for 3 days. To activate the YAP/TAZ signalling cascade, LATS-kinase inhibitor (N-(3-benzylthiazol-2(3H)-ylidene)-1H-pyrrolo[2,3-b]pyridine-3-carboxamide, LATSi) (MedChemExpress, Monmouth Junction, NJ, USA) at a concentration of 10 µM, was added for 5 days.

### 2.3. Flow Cytometry

The cell pellet was fixed in 4% paraformaldehyde (PFA) solution (Sigma-Aldrich, St. Louis, MO, USA), then resuspended in PBS (PanEco, Russia) with 10% FBS for further staining with Primary rabbit anti-KRT15 (Abcam, Waltham, MA, USA, ab52816, dilution 1:100) and Primary rabbit anti-Col17a1 (Novus Biologicals, Littleton, CO, USA, NBP2-38686, dilution 1:100) for 1 h. Then, the cells were rinsed in PBS, and secondary donkey anti-rabbit Alexa Fluor 647 (1:800, Abcam) antibodies were added and incubated for 1 h. Flow cytometry was performed with an Attune NxT flow cytometer (Thermo Fisher Scientific, Waltham, MA, USA). Each experiment was repeated three times.

### 2.4. Immunohistochemical Staining and Imaging

Cryosections were sliced to a thickness of 7 μm using a Leica CM1950 cryostat (Leica Biosystems, Nussloch, Germany).

For haematoxylin–eosin (H&E) staining sections were fixed in 4% of PFA solution (Sigma-Aldrich, USA), rinsed in DPBS, and incubated with haematoxylin (BioVitrum, Moscow, Russia) for 3 min, rinsed in tap water for 10 min, then incubated in eosin solution (BioVitrum, Russia) for 5 s, washed in distilled water, and embedded into a Bio Mount medium (BioOptica, Milano, Italy). H&E-stained slides were imaged with a Keyence BZ-9000 microscope (Keyence, Osaka, Japan).

For BrdU-Ki67 staining, sections were fixed in cold 70% ethanol, followed by DNA hydrolysis in 4N HCl mixed with 70% ethanol (1:1). After epitope retrieval in 0.0125% trypsin, sections were incubated in 1% H_2_O_2_.

For the remaining antibody panel staining, sections were fixed in 4% of paraformaldehyde (PFA) solution (Sigma-Aldrich, USA).

For following immunohistochemical (IHC) staining, sections were incubated with primary antibodies (Table 1) diluted in DPBS solution (PanEco, Russia) containing 5% BSA (PAA, Pasching, Austria), 1% Triton X-100 (AppliChem, Darmstadt, Germany), and 1% Tween-20 (AppliChem, Germany) for 18 h at 4 °C. Then, the sections were rinsed in DPBS and incubated with secondary antibodies (Table 1) for 1.5 h. Nuclei were additionally stained with DAPI solution (Biotium, Fremont, CA, USA) for 10 min. Images were captured with a Leica Thunder microscope with THUNDER Imager Tissue for resolution improvement (Leica Microsystems, Wetzlar, Germany).

### 2.5. Image Analysis

The immunohistochemical images of BrdU and Ki67 co-staining were imaged with Leica Thunder using THUNDER Imager Tissue LAS X software version 3.0.0.15697 (Leica Microsystems, Wetzlar, Germany) for background signal reduction. The high-resolution images (Figure 1a,b) were then analysed in QuPath-0.5.0 software [32]. A total of 3–5 tissue sections for each biopsy have been analysed (a panorama of epidermal region including 5–36 fields of view). Epidermis borders, basal membrane, and epidermis surface, as well as HFs were manually detected. StarDist extension for QuPath has been applied to detect nuclei in epidermis based on DAPI staining [33]. As Ki67 and BrdU positive nuclei were located mainly in lower epidermal layers and due to high background signal in upper epidermal layers (Figure 1c, white arrow) which could be detected as signal from BrdU or Ki67, nuclei that located in upper epidermal layers were excluded from analysis. Nuclei intensity features (Min, Median, Mean, SD, and Max) were measured for all channels. BrdU/Ki67-positive nuclei were detected by intensity-based nuclei thresholding in corresponding channel (Figure 1d, blue objects).

For each nuclei a set of parameters was measured in QuPath software: nuclei area, circularity, solidity, maximal diameter, minimal diameter, Ki67/BrdU/DAPI mean, median, minimal and maximal intensity, distance to the annotations with basal membrane and epidermis surface, distance to the HF. In order to ensure the most accurate and informative results, we avoided using measurements that reflected similar morphological changes and selected the most informative ones. In this case, we dropped Ki67/BrdU/DAPI median, minimal, and maximal intensities as mean fluorescence intensity for Ki67, BrdU, and DAPI provides a more comprehensive insight. Additionally, the maximal and minimal diameters of the nuclei were excluded from the analysis, as the nuclei area and circularity correlated with these parameters but provided more informative results. We also avoided using solidity in our analysis. This parameter reflects the curvature of the object border. However, the integrity of fluorescence signal at nuclei borders could be affected by the detection issues.

### 2.6. Statistical and Cluster Analysis

Data analysis was carried out using Python 3.9.20. For descriptive statistics, prior to selecting an appropriate statistical criterion, each dataset was analysed for normality of distribution using the Shapiro–Wilk test, Kolmogorov–Smirnov test, D’Agostino–Pearson combined test, and Anderson–Darling test. For normally distributed datasets, a one-way analysis of variance (ANOVA) with the post hoc Tukey test for multiple comparisons was used. For the data that deviated from normal distribution, differences were assessed with the Kruskal–Wallis test and the post hoc Dunn’s test for multiple comparisons. The presence of outliners in the data was determined with the help of Dixon’s Q test.

K-means clustering was performed to identify keratinocyte subpopulations based on their morphology. Ki67 and BrdU mean intensities, nuclei area, and nuclei circularity were used as input parameters for clustering. The optimal number of clusters was determined using the Elbow Method and Silhouette Method.

### 2.7. Single Cell Data Analysis

Raw count matrices were mined from the Gene Expression Omnibus (GEO) database under accession GSE147482, originally published in [15]. The data were processed and analysed using the R package Seurat (v4.3.0), following the processing performed in the original study. Additionally, cell cycle analysis was performed using the R package tricycle [34].

## 3. Results

### 3.1. Regeneration of Human Skin Xenograft

The human skin xenograft completely integrated into mouse skin by PG110. Both head and abdominal xenografts restored the native skin architecture and contained all skin layers and derivates, including epidermis, dermis, subcutaneous fat, and sweat glands. The head xenografts also contained HFs with sebaceous glands (Figure 2). The epidermis possessed key histologic features of normal human skin, including basal and suprabasal marker expression (Figure 2g,h) and rete ridge formation, although it was slightly thicker compared to intact skin. The dermis contained well-distinguished papillary and reticular layers [30]. Dermal papillae surrounded epidermal rete ridges (Figure 2c,d), thereby contributing to the formation of epidermal topology. In the head xenograft, HFs were attached to the arrector pili muscle (Figure 2e,f) and sebaceous glands (Figure 2a,b). They reached the anagen stage of the hair cycle and produced hair shafts.

As the majority of parameters returned to their normal state, the epidermis in the xenograft at PG110 was considered a model of normal skin IFE.

### 3.2. Clustering of Proliferative Cells in Epidermal Basal Layer

In order to study the distribution of proliferating cells in the skin xenograft, we applied QuPath software to identify Ki67+/BrdU+ cells and then create density maps of cell distribution, and then we further measured the spatial characteristics of the identified cells—distance from HFs and epidermal thickness for each cell type (Appendix A). Epidermal thickness was calculated as the sum of distances to the basement membrane and to the epidermal surface for each nucleus; thereby, nuclei situated in regions of greater epidermal thickness were more likely to be located in rete ridges.

BrdU and Ki67 intensities, as well as morphological parameters (area and circularity), were used to perform k-means clustering for nuclei of basal proliferating cells. The selection of clustering parameters is described in more detail in the Appendix A. Briefly, Ki67 intensity reflected the cell cycle state, while BrdU+ cells included proliferative label. BrdU intensity further gave us an information about the number of divisions: cells with lower BrdU content undergone more divisions and diluted label. BrdU/Ki67-negative cells were excluded from the analysis to prevent misinterpretation of the results. Epidermal thickness and distance to the HF were not included in the clustering parameters to avoid artificially clustering of populations based on their position in the epidermis, but further these parameters were counted for each cell cluster to verify whether distinct populations with particular morphological characteristics can occupy specific locations. Incorporating spatial proximity directly into the k-means clustering algorithm would have risked artificially grouping cells based solely on their location, rather than their inherent cellular features.

The first step of clustering is projecting all the input parameters onto a 2D map. Each object (i.e., nuclei) achieves its spatial position based on the measured values. The next step involved mapping centroids for each cluster based on the pre-defined number of clusters and assigning a cluster to each object. Centroid mapping and cluster assignment were repeated until the objects in each cluster achieved the minimal distance to the cluster centroid. To determine the optimal number of clusters, we utilised the Elbow Method and Silhouette Method. In our analysis, we identified that dividing objects into four clusters resulted in a meaningful set of clusters (Figure 3a,c).

To assess the spatial distribution related to rete ridges, epidermal thickness was calculated for each cluster (Figure 3b,d). Additionally, a scatter plot was provided with a graph of epidermal thickness versus BrdU intensity for the examination of potential correlations (Figure 3e). A unique number was assigned to each nucleus detected in QuPath. Following cluster analysis, the object numbers were employed to identify the nuclei in the initial image. Consequently, we were able to depict the clusters in the epidermis (Figure 3f).

We first concentrated on clusters identified in head xenografts. All the clusters possessed their own unique features. Cluster 3 contained nuclei with both Ki67 and BrdU maximal staining intensities, including a group of cells with rare divisions based on BrdU staining (Appendix A). The mean epidermal thickness parameter for these cells was not significantly different from those of Clusters 1 and 2 (Figure 3b). However, mapping cell clusters at scatter plot revealed that cells with the highest BrdU intensity were predominantly located in the areas with the lowest epidermal thickness (Figure 3c), i.e., inter-ridge regions.

In contrast, cells in Cluster 4 were concentrated in areas with maximal epidermal thickness (Figure 3b), indicating localization in rete ridges. Another key feature was the larger nuclear area compared to other clusters (Figure 3b).

Clusters 1 and 2 exhibited similar distributions in terms of BrdU/Ki67 intensity and epidermal thickness but differed in nuclear shape. The cells in Cluster 2 had more elongated nuclei with a smaller area. Morphological characteristics of nuclei in Cluster 1 were similar to those observed in Clusters 3 and 4.

Reviewing spatial distribution, cells from Clusters 1 and 2 were uniformly spread throughout the epidermis; Cluster 4 was mostly located in rete ridges, while Cluster 3 was found in inter-ridge areas.

Clusters found in abdominal xenografts slightly differed from those in the head region but possessed some common morphological characteristics. The epidermal relief in the abdominal skin was less pronounced compared to head skin; however, the differences between clusters were observed. Clusters 1 and 2 were similar to that observed in the head xenograft. Cluster 3 contained rarely dividing nuclei with high BrdU but low Ki67 intensity. Epidermal thickness for Cluster 3 was low, indicating its localization in inter-ridge regions. High Ki67 staining was observed in cells of Cluster 4, which predominantly occupied the rete ridges (Figure 3d).

Cluster features are summarised in Table 2.

### 3.3. HFs Influence on Proliferating Cell Distribution

A detailed analysis was conducted to verify the hypothesis that the distance from the cell to the HF may influence cell proliferation activity [35]. We did not find any correlation between HF localization and Ki67/BrdU staining (Appendix A). However, grouping by BrdU label intensity has shown an increased distance from the HFs for rarely dividing cells (Appendix A). For further analysis, we created plots describing the distribution of BrdU intensity against the distance to HFs (Figure 4a,b). It demonstrates that rapidly dividing cells with low BrdU levels were concentrated within 100 µm from HFs, while cells with maximal BrdU appeared at a distance greater than 750 µm (Figure 4a,b). We grouped cells by their distance from HF (Figure 4c) and then analszed them by the intensity of BrdU staining. Cells located within 100 µm around HFs had minimal BrdU intensity level compared to other groups (Figure 4e). No cells from Cluster 3 were observed in this region, while an increase in the percentage of cells from Cluster 1 was detected (Figure 4d). Cells from Cluster 3 exhibited a tendency to localise in more remote regions relative to the HF, while cells from Cluster 4 demonstrated a propensity to congregate in closer proximity to the HF (Figure 4g).

### 3.4. Xenograft Regeneration

Human skin gradually restored its structure after transplantation into immunodeficient mice. At PG40, the dermis was unstructured, sweat glands were absent, and HFs were in a wound-induced telogen phase. At PG75, papillary dermis appeared beneath the epidermis, and HFs entered the growth phase [30]. The epidermis showed hypertrophy at PG40, but its thickness decreased during the regeneration process (Figure 5).

The rete ridges in the epidermis were barely distinguishable at PG40. By PG75, large areas of flat epidermis were intermixed with invaginations, and by PG110, rete ridges became evident (Figure 5a–f and Figure 6). To trace the restoration of rete ridges, we studied the distribution of keratin 15 (KRT15) and collagen XVIIa (Col17a1), which are considered markers of cells located in rete ridges and inter-ridges, respectively.

The fluorescence intensity of KRT15 and Col17a1 was analysed in the cells of the basal epidermal layer (Figure 6a,b). The cells were categorised according to their position in the invaginations or evaginations, based on the thickness of the epidermis at the location of the cell. The fluorescence intensity was then compared between the invaginations and evaginations. Firstly, we examined the distribution of the markers in intact skin. The intensity of KRT15 was higher in invaginations that corresponded to rete ridges, while Col17a1 was upregulated in evaginations (inter-ridges).

In the initial stages of xenograft regeneration, KRT15/Col17a1 staining was uniformly distributed within the epidermis and had no specific pattern. The restoration of epidermal relief began at PG75 (Figure 6a) and became sustainable by PG110, with distinct separation of KRT15 and Col17a1 areas.

### 3.5. Distribution of Cell Clusters During Xenograft Regeneration

The dynamics of cell proliferation and spatial distribution was traced during xenograft regeneration. In the head xenograft at PG40, the percentage of Ki67+ cells in the basal layer was higher than at PG75 and PG110 (Figure 7a,b). All the identified cell clusters persisted. Cluster 4 tended to be less abundant at PG40 and PG75, while at these stages, the number of cells from Cluster 1 slightly increased (Figure 7c). In abdominal skin at PG40 and PG75, only a few BrdU+ cells were observed, while their amount restored by the PG110 (Figure 7a). In contrast to head skin, the expression of Ki67 remained high at all stages of xenograft regeneration showing no correlation with BrdU label incorporation.

From PG40 to PG75, the mean epidermal thickness decreased in both types of xenografts for all clusters, corresponding to an overall reduction in epidermal thickness (Figure 5b). This trend continued up to PG110, except for Cluster 4, (Figure 7e) where crowding of cells in rete ridges was observed (Figure 7e). Cells from Cluster 3 in head skin were also uniformly distributed at early stages of regeneration regardless of the epidermal thickness, and similarly these cells concentrated in inter-ridges at PG110 (Figure 7d). In abdominal skin, Cluster 3 was almost absent at PG40 and PG75 but restored by PG110 predominantly occupying the inter-ridge region (Figure 7d,e). While proliferating cell clusters reached their specific localization only by PG110, the formation of rete ridges, as shown shown by the Col17a1 and KRT15 pattern (Figure 6a,b), initiated at an earlier stage.

The HF-dependent spatial distribution in the head xenograft appeared earlier during xenograft regeneration. At PG40 and PG75, cells with varying BrdU intensities were uniformly distributed around the HFs (Figure 8a); however, by PG75, a concentration of proliferating cells was observed within a 750 µm radius around the HFs. The correlation between proliferation activity and distance from the HFs was consistent across all clusters, with an increase in “Distance to HF” corresponding to an increase in xenograft size (Figure 8b). Within a 100 µm radius of the HF, proliferation intensity increased, as indicated by a decrease in mean BrdU intensity (Figure 8c).

### 3.6. Cluster Definition

Our next goal was to characterise the cell states inside the clusters that were identified. We analysed the expression of several epigenetic markers and transcription factors in the abdominal skin xenograft (Figure 9a). We stained di-/tri-methylated H4K20 (H4K20me2/3), SETD8, or KLF4 in combination with BrdU and Ki67 markers in order to trace the expression of these markers in the described clusters. We found elevated levels of H4K20me2/3 in combination with an increase in SETD8 expression in Cluster 4, located within rete ridges (Figure 9b). Tri-methylation at histone H4 is changed to mono- and di-methylation upon the onset of terminal differentiation. Mono-methylation is maintained by SETD8 methyltransferase [36]. SETD8 expression is crucial for c-Myc-induced keratinocyte proliferation and differentiation [37]. The active proliferation of cells from Cluster 4 may be consistent with SETD8 ability to maintain epidermal proliferation [38]. Based on these observations, it is assumed that Cluster 4 contains progenitor cells that exited stem cell compartment.

We next studied the expression of KLF4, which is considered as a master-gene of epidermal differentiation. KLF4 is thought to encourage the differentiation of epidermal cells; its suppression preserves the stem and progenitor status of keratinocytes [39,40]. We observed decreased KLF4 expression in epidermal Clusters 2 and 4 (Figure 9b). While Cluster 2 expresses low levels of SETD8 compared to Cluster 4, low expression of KLF4 in combination with small, elongated nuclei indicates a less differentiated state. KLF4 is often considered to interact with the YAP/TAZ-TEAD complex mediating keratinocyte differentiation. Basically, YAP antagonises KLF4 to preserve undifferentiated cell state and promote proliferation [41].

We therefore aimed to analyse the interaction between YAP and KLF4 and their influence on proliferative pattern formation. We observed a significant correlation between KLF4 staining intensity and YAP activity, with a low correlation to Ki67 expression in the epidermis (Figure 9c), indicating their cooperation in basal cell states patterning and proliferation control.

Next, we investigated YAP activity and KLF4 expression during epidermal regeneration (Figure 10a). The expression of both KLF4 and YAP was elevated at PG40. Active nuclear YAP was detected not only in the basal layer of the epidermis but also in individual keratinocytes in suprabasal layers. The expression of KLF4 returned to the normal state by PG75, while YAP expression gradually decreased and achieved a minimum at PG110 (Figure 10a,b). KLF4 is able to upregulate the expression of YAP [41], and both factors may be crucial for the early regeneration stages. In the later stages of regeneration, activated YAP may induce hypertrophy and hyperproliferation of the epidermis [42,43,44].

### 3.7. YAP Influence on Cell Proliferation Dynamics

In order to verify the hypothesis that YAP activity exerts an influence on the intensity of proliferation, the distribution of proliferative cells was studied in primary human keratinocytes in normal state and after YAP activation using LATS kinase inhibitor (Figure 11). Inhibition of LATS resulted in the nuclear translocation of YAP, inducing its activated state. We conducted a short-term BrdU pulse test to investigate the influence of YAP activation on cell proliferation.

Our results showed an increase inproliferation, as evidenced by enhanced BrdU and Ki67 intensity. However, KLF4 expression slightly decreased compared to the control group (Figure 11a,b).

YAP was found as one of key factors involved in the formation of rete ridges [45]. We analysed the influence of YAP activation on the expression of Col17a1 and KRT15, markers associated with rete ridge formation. LATSi treatment induces the reduction in Col17a1 fluorescence intensity and an increase in KRT15 intensity, as observed through IHC (Figure 12a,c). We further performed flow cytometry analysis to investigate Col17a1 and KRT15 expression changes more precisely. We studied the distribution of negative, dim, and bright cells. All keratinocytes expressed Col17a1. In the presence of LATSi, the amount of bright fraction reduced while the fraction of dim cells increased. Regarding KRT15, KRT15-negative fraction was observed. In the presence of LATSi, the fraction of KRT15-negative cells reduced; however, the mean intensity of KRT15 staining decreased as well as Col17a (Figure 12b,d).

### 3.8. Transcriptional Differences in Proliferating Cell Populations

Next, we checked whether the epidermal clusters observed in the xenograft could be identified in normal human skin. We re-analysed the single-cell sequencing dataset of human skin. We concentrated on the study of four basal epidermal cell clusters identified by Wang et al. (GSE147482, originally published in [15]) (Figure 13a,b). In the original study, the differentiation trajectory started from a single cluster of stem cells and then moved to cells committed to differentiation in rete ridges. A COL17A1-high cluster located in the inter-ridges was also observed [15]. An alternative study based on the same dataset showed the opposite pattern, identifying two populations of stem cells which differed by the expression of LRC and non-LRC signatures differentiated into the single population of basal cells which further started terminal differentiation [8]. We next aimed to trace the transcriptional patterns corresponding the cell clusters in the skin xenograft model. We filtered the dataset and further analysed only the populations of basal keratinocytes.

We started with the analysis of the cell cycle in the basal populations. The analysis shown that the cell cycle in the studied populations was synchronized, showing little fluctuations (Figure 13c). This synchronization allows the utilization of Ki67 as one of the markers to identify cell populations based in IHC staining, while differences in cell cycle stages among cells could introduce fluctuations in Ki67 staining intensity.

The Basal III cluster was characterised by high COL17A1 levels being located at inter-ridges. Low proliferation activity, confirmed by low expression of MKI67 and CCND1, indicated that these cells may correspond to BrdU-retaining cells in Cluster 3 of the xenograft, which were also located in the inter-ridge region. These cells were characterised by the active YAP-target expression (CYR61, SNAI2) (Figure 13d). The decreased proliferation rate, in combination with activated YAP target expression, indicates that YAP in this context may primarily participate in epidermal relief establishment.

Basal I and II clusters are characterised by the highest expression of proliferation marker MKI67, which are also associated with a less differentiated phenotype. The lowest level of KLF4 in Basal II cluster corresponds to the low expression of KLF4 protein observed in Cluster 2 of our experiments. Basal I is characterised by active proliferation as well es high levels of KRT15 expression. Moreover, KMT5A (corresponding to SETD8) is upregulated in Basal I, indicating a resemblance between Basal I and Cluster 4 (Figure 13d). Basal IV shares similarities with other clusters and may be referred to as an intermediate state, resembling our Cluster 1.

## 4. Discussion

The application of a designed algorithm based on QuPath software enabled the description of proliferative pattern in human epidermis. Initially, the parameters that could be extracted from histological images were analysed. Skin sections were stained for BrdU and Ki67, which mark proliferating cells. It was also demonstrated that nuclear shape can be used as a parameter reflecting the proliferative state of the cell (Appendix A). The application of these parameters enabled the isolation of four clusters of proliferative cells in human skin IFE (Table 2). Clusters 1 and 2 were identified throughout the epidermis, while cells from Clusters 3 and 4 exhibited a tendency to concentrate in specific locations, namely inter-ridges and rete ridges, respectively (Figure 3). Rete ridges have been previously described as a putative niche for epidermal stem cell localization [13,14].

The highest Ki67 level was observed for Cluster 3 in head xenografts and for Cluster 4 in abdominal skin (Figure 3b,d). Ki67 expression undergoes a gradual change during different phases of the cell cycle, reaching a maximum in the M phase and a minimum in G1 [46,47]. This indicates that cells from Cluster 3 and 4 are actively cycling and differ from the others in the phase of the cell cycle. A comparison of the cell nuclei features (Table 2) indicates that cells belonging to Cluster 3 in the inter-ridges underwent a reduced number of cell divisions, as evidenced by the accumulation of BrdU. The most notable feature of cells in Cluster 4 is their localization in rete ridges and relatively large nuclear area (Figure 3b, Table 2).

While the dependence of cellular proliferation patterns on rete ridges is widely discussed [8,48,49,50], discrepancies exist in the assumptions concerning the stemness of cells associated with rete and inter-ridges. Some studies consider two stem cell populations with different division rates [8], while another hypothesis suggests these populations are transit-amplifying cells originating from a single stem population [15]. We therefore questioned whether the identified populations possess stem or differentiation properties. We identified an elevated level of SETD8 methyltransferase in Cluster 4 (Figure 9b). The analysis of single-cell data has shown that KRT15^high^ cluster of basal epidermal cells, expressing high levels of proliferation markers, had an elevated KMT5A (SETD8) expression, resembling the identified Cluster 4 in our model (Figure 13d). While SETD8 mediates mono-metlylation of histone H4 at lysine 20 and the onset of keratinocyte differentiation [36], Cluster 4 may be referred to as transit-amplifying cells that have started differentiation.

The other two clusters have low levels of BrdU and Ki67 and exhibit only a variation in nuclear shape, with those of Cluster 2 being more elongated. The elongated nuclei with a lower area are characteristic of the epidermal basal layer in comparison to cells that have undergone superficial differentiation [51,52]. This observation was corroborated by our data (Appendix A). Cells from Cluster 2 expressed low levels of KLF4 (Figure 9b), indicating a less differentiated state, which should correspond to epidermal stem cells [53]. The observation that stem cells are distributed throughout the basal epidermal layer aligns with a previous study which employed lineage tracing on the human skin xenograft, which showed no preference for relief in the distribution of stem cells [29].

Cluster 1 had a mixed phenotype and could contain keratinocytes in a transitional state exiting stem compartment. Cells from Cluster 2, after the onset of differentiation, move towards Cluster 1 and then acquire the differentiated phenotype.

HFs and rete ridges are considered to be organisers of the spatial patterning of proliferation in IFE. In a xenograft model comprising both HFs and extensive regions of inner follicular epithelium (IFE), we are able to examine the relationship between proliferation dynamics and distance to HFs. The results demonstrated that rapidly proliferative cells are concentrated in areas surrounding the HF infundibulum, while slow-cycling cells appeared at a distance greater than 750 µm (Figure 4). These cells with high proliferation intensity were identified as belonging predominantly to Cluster 1, indicating that they are amplifying progenitors rather than stem cells (Figure 4a,d). A similar proliferation pattern was observed in mouse epidermis [35].

During the process of skin regeneration in xenografts, the distribution of epidermal cells undergoes rearrangement and subsequently assumes the structural characteristics of homeostatic skin. This is accompanied by a reduction in proliferation, epidermal thickness, re-entry of HFs into the growth phase, and recovery of epidermal undulated relief [17,18,30,31,54]. The restoration of epidermal proliferation patterns was subjected to analysis. In the head xenograft it was surprising to find that cells belonging to all four clusters were present at PG40. The elevated proliferation observed in the early stages of regeneration is not attributable to a single cluster. Furthermore, the ratio between clusters remains constant from PG40 to PG110 (Figure 7c). At PG40, cells from all four clusters are observed to uniformly spread across the epidermis. At PG75, the accumulation of proliferative cells around HFs is noted, with the complete proliferation pattern observed only at PG110.

The process of rete ridge formation is accompanied by the specific positioning of markers [8,55] (Figure 6) and papillary dermis restoration [30]. In the initial stages of xenograft regeneration, these domains are indistinguishable, and thus epidermal relief does not contribute to proliferation. Following the regeneration of the rete ridge, the cells belonging to Clusters 3 and 4 re-occupy their own specific territories in both types of the xenograft, but they do not appear in the specific cell niches in response to the induction signals. The distribution of KRT15 and Col17a1 expression reorganises during regeneration in a manner that is consistent with the spreading of the cells belonging to Clusters 3 and 4 (Figure 6 and Figure 7). In contrast to the head skin, Cluster 3 was not identified in abdominal xenograft until PG110 (Figure 7d,e). The expression of Ki67 remained the same during regeneration (Figure 7a), indicating different cellular dynamics from that observed in head skin.

YAP, in conjunction with KLF, is responsible for the regulation of the processes of differentiation, proliferation, regeneration, and formation of skin relief [41,56]. The reduction in Col17a1 and KRT15 expression in cells with activated YAP, in combination with active YAP at early regeneration stages, contributes to the hypothesis that hyperactivated YAP prevents proper relief formation. The spatial allocation of Clusters 3 and 4 occurs after the eventual restoration of relief and the decrease in YAP activity. Based on the single-cell data, Cluster 4 corresponds to Basal I with minimal YAP activity.

In the initial phases of skin regeneration, the role of YAP is considered to be of significance in the process of skin restoration [57,58], supporting active proliferation [59] and migration [60,61] and preventing apoptosis [62]. However, it has been demonstrated that persistent activity of YAP results in the induction of epithelial–mesenchymal transition [63,64], an increase in proinflammatory cytokine and growth factor levels [65,66,67,68], and the onset of tumours [45]. Our data indicate the important role of YAP in the control of proliferation and formation of the epidermal relief, as well as in spatially distinct cell clusters. The activity of KLF4 is also important for the proper regeneration process [69]. While YAP has been shown to negatively regulate KLF4 expression [41], the overexpression of KLF4 may induce YAP activation during regeneration [70]. The gradual decline in KLF4 expression during xenograft regeneration, concomitant with the decrease in YAP activity (Figure 10), is a prerequisite for normal epidermal differentiation.

Our results demonstrate the distinctive patterning of cell proliferation states in the IFE in relation to the rete ridges and HFs. The role of rete ridges in the regulation of proliferation kinetics remains a topic of debate [8,29,48]. However, the formation of rete ridges has been shown to enhance the bio-mechanical properties of engineered skin in 3D equivalents [49,71]. Additionally, the reduction in rete ridge height has been associated with the aging process [72]. Our data indicate the difference in proliferation patterns for skin in different body regions. The main difference between head and abdominal skin is the abundance of HFs that may influence the regeneration dynamics. It is established that HFs provide epidermal cells during mouse skin regeneration [73]. Their appearance in a wound has been demonstrated to prevent scar formation and result in complete skin structure reconstruction [74]. In the context of medical practice, the transplantation of HFs is regarded as a technique that can facilitate wound closure [75] and reduce the formation of scars [76]. Our data indicates that HFs may serve to normalise not only dermal structure but also epidermal proliferation patterns.

A number of dermatological conditions are linked to abnormalities in cell proliferation kinetics. An understanding of normal physiological processes is paramount to elucidating the pathogenesis of the disease. In this context, our results highlighting the critical role of YAP in maintaining epidermal homeostasis are particularly significant. The precise regulation of YAP activity is essential for proper epidermal function, and disruptions to this regulation are increasingly implicated in various skin pathologies. The hyperactivation of YAP is associated with psoriasis [77,78], tumours such as squamous cell carcinoma [64,79,80], and basal cell carcinoma [81,82,83], while YAP deficiency is often associated with epidermolysis bullosa [84] or may prevent chronic wounds from healing in conditions such as epidermolysis, diabetes, and ageing [57]. Therefore, the insights gained from our study, demonstrating contribution of YAP to epidermal homeostasis under normal conditions, provide a crucial framework for interpreting the alterations in YAP activity observed in diseased states. The identification of the characteristics that underpin epidermal homeostasis enables the formulation of strategies for reestablishing the skin’s normal state in pathological conditions. Furthermore, xenotransplantation of human skin opens up broad prospects for disease modelling [16]. A range of models have been developed, including those of acute wounds [85,86,87], pressure ulcers [21,88], psoriasis [89,90], and viral infections [20,22]. These models are intended to facilitate the subsequent study of treatment options identified in the model of normal regeneration.

In the field of human epidermal research, priority approaches for study include single-cell analysis, with subsequent identification of IHC populations. While single-cell RNA sequencing provides valuable insights into cellular heterogeneity, it inherently lacks spatial context, precluding direct assessment of cell localization within the tissue architecture. Immunohistochemical staining could give information about spatial distribution of cell clusters; however, it may not accurately reflect patterns identified by single-cell data analysis due to post-translational regulatory mechanisms. Furthermore, while spatial transcriptomics offers the potential to bridge this gap, current technologies lack the resolution necessary to resolve the subtle, yet critical, spatial relationships of cell populations within the complex microenvironment of the epidermis. IHC staining of human skin biopsies does not reflect the actual proliferation pattern, as Ki67 and other proliferation markers do not allow tracking of dynamical changes. In this case, human skin xenotransplantation allows us to capture the spatial dynamics of proliferation more comprehensively.

The utilisation of human skin xenografts facilitated the identification of stem and progenitor cell populations within the epidermis. While many studies consider the xenograft model to be the norm, some aspects of the model should be taken into account when interpreting the results. These include the infiltration of human grafts with mouse cells [91], the participation of host cells in regeneration processes [92], as well as neovascularization which occurs with the help of mouse endothelial cells [93], and the loss of human resident immune cells [94]. The present study demonstrated that the skin xenograft model exhibited activated YAP signalling even at PG110, which may be the underlying cause of the thickened epidermis observed in comparison to normal skin. These findings suggest that partial inhibition of YAP activity in future xenograft studies could potentially lead to the restoration of a normal phenotype. The utilisation of different skin sources enabled the identification of analogous cell populations, thereby indicating the persistence of cell types even within a system that excluded mouse cells. It is also worth noting that the regenerative potential of xenografted skin is significantly influenced by factors such as donor age and individual characteristics, even in cases involving the use of skin from healthy donors undergoing cosmetic surgeries. To ensure the attainment of reproducible results, it is imperative to select donors of similar age and without underlying pathologies. The investigation of age-related disparities in skin regeneration following xenotransplantation and the assessment of the influence of pathological conditions on this process merit further exploration in future studies.

The combination of in vivo labelling followed by computational analysis with high-resolution imaging allowed the identification of several clusters of proliferating cells in the epidermis. The precise hypothesis regarding epidermal stem cell organisation remains uncertain; it encompasses the hypothesis of two stem cell populations [8] and the hypothesis of a single progenitor [15], both of which have gained support. The obtained data allow us to propose that the single progenitor hypothesis [5] is largely consistent with our results. The single cluster corresponding to the less differentiated cells, which could be referred to as stem cells, is uniformly distributed in the epidermis. The clusters that were identified as being associated with the epidermal relief possess features of more differentiated cells, thus matching the described differentiation trajectories from the single stem cluster to spatially distinct populations [15]. The hypothesis of two distinct stem cell populations with different proliferation rates [8] is rendered less probable due to the differentiation status of the spatially associated clusters. Furthermore, this finding aligns with another study performed on a xenograft model, which observed no spatial proximity of proliferating clones [29]. Consequently, the present findings suggest the necessity for further investigation into the mechanisms that govern the formation of proliferation hierarchies, given that a single stem-like population gives rise to spatially segregated populations exhibiting differential proliferation rates.

## 5. Conclusions

A computational image analysis of the proliferation kinetics in human skin xenografts has identified four clusters related to different proliferation states. The single cluster of stem cells and the cluster of cells in transitional state are dispersed throughout the epidermis, while the two remaining clusters are referred to as progenitors situated in disparate territories. Rapidly cycling progenitors are located in rete ridges, whereas slow-cycling cells are situated in inter-ridge areas. Cells undergoing active proliferation are concentrated in the vicinity of HFs. All cell states are present during skin regeneration. They repopulate specific territories following the recovery of epidermal undulation. This process occurs in response to a decrease in YAP signalling activity.

## Figures and Tables

**Figure 1 cells-14-00448-f001:**
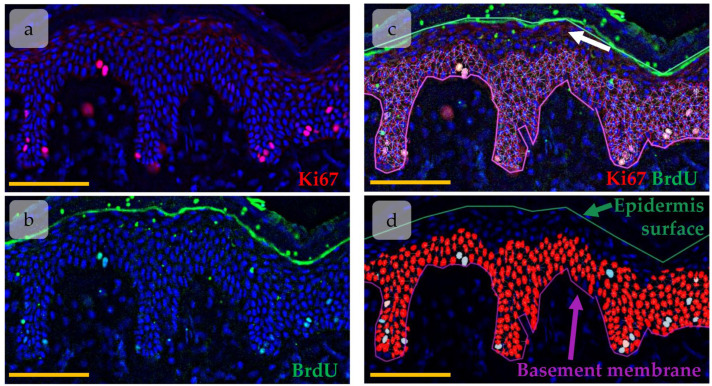
Object detection in epidermis using QuPath software. (**a**,**b**) Immunohistochemical (IHC) staining of Ki67 (**a**) and BrdU (**b**). Ki67 (red) and BrdU (green) are overlaid onto DAPI (blue). (**c**,**d**) Nuclei detected with the StarDist extension (red). White arrow—background signal in upper epidermal layers. Scale bar—100 µm.

**Figure 2 cells-14-00448-f002:**
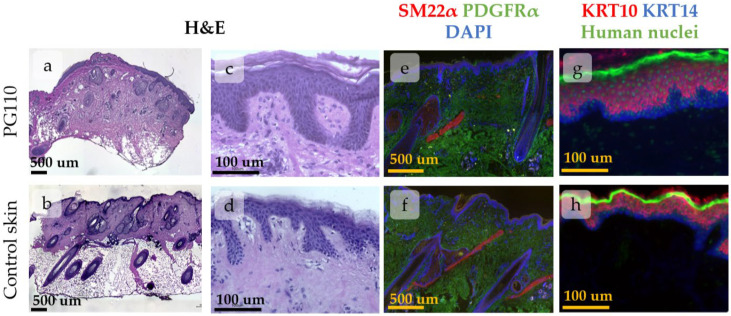
Histological images of xenografted skin at PG110 compared to the control tissue, showing the full restoration of tissue structure. Scale bar—500 µm or 100 µm. (**a**–**d**) Haematoxylin–eosin staining of xenografted skin with the hair follicles (HFs) and sebaceous glands (**a**,**b**). Whole-section images (**a**,**b**) and an enlarged view of the epidermis with the formed rete ridges (**c**,**d**). (**e**–**h**) IHC staining for specific markers. (**e**,**f**) Staining for SM22α (arrector pili muscles) and PDGFRα (total dermal fibroblasts). Nuclei are stained with DAPI. (**g**,**h**) Staining for KRT10 (keratinocytes at suprabasal skin layers) and KRT14 (keratinocytes at basal skin layer). Nuclei are stained with anti-human nuclei antibodies.

**Figure 3 cells-14-00448-f003:**
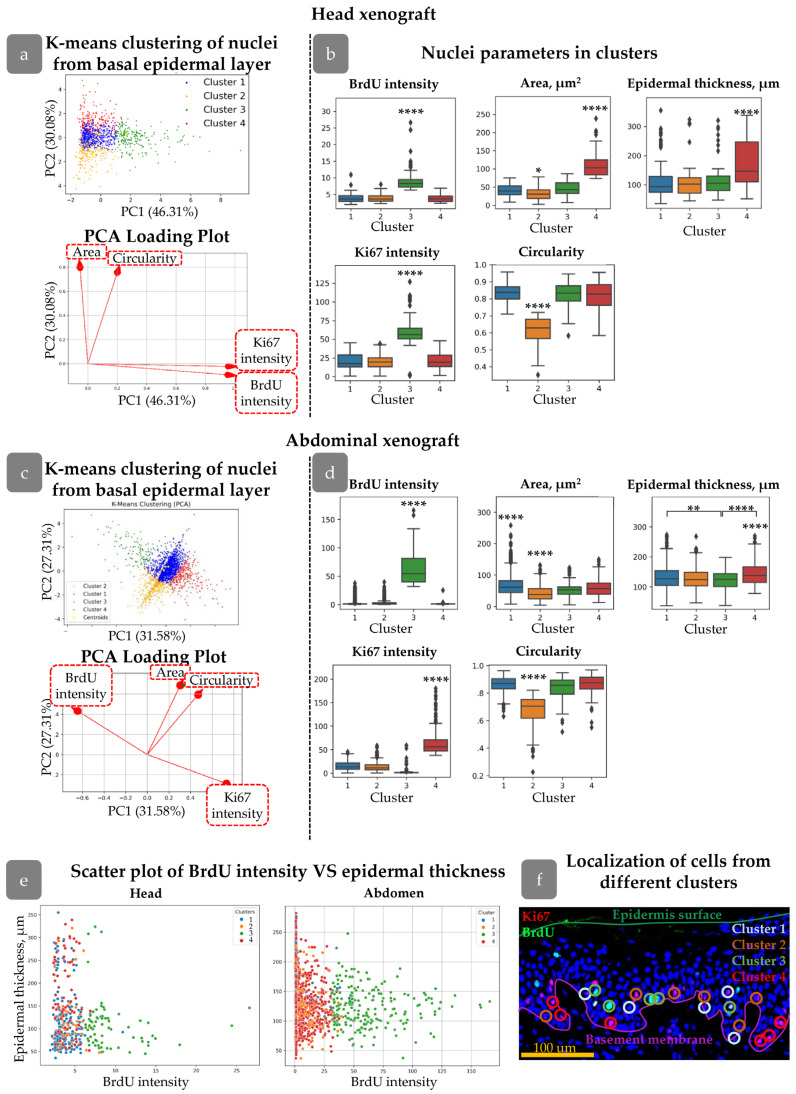
Clustering of Ki67+/BrdU+ nuclei in head (**a**,**b**) and abdominal (**c**,**d**) xenografts. (**a**,**c**) Principal component analysis (PCA) and PCA loading plots of analysed cells showing four clusters in the PC coordinates. (**b**,**d**) Morphological parameters of nuclei from four clusters. Outliers are shown on graph. **—*p*<0.01, ****—*p* < 0.0001 relative to other clusters (one-way ANOVA with Tukey correction for multiple comparisons). (**e**) Scatter plots of BrdU intensity versus epidermal thickness for cells from different clusters. (**f**) Examples of cells from different clusters in skin stained with BrdU (green), Ki67 (red), and DAPI (blue). Scale bar: 100 µm.

**Figure 4 cells-14-00448-f004:**
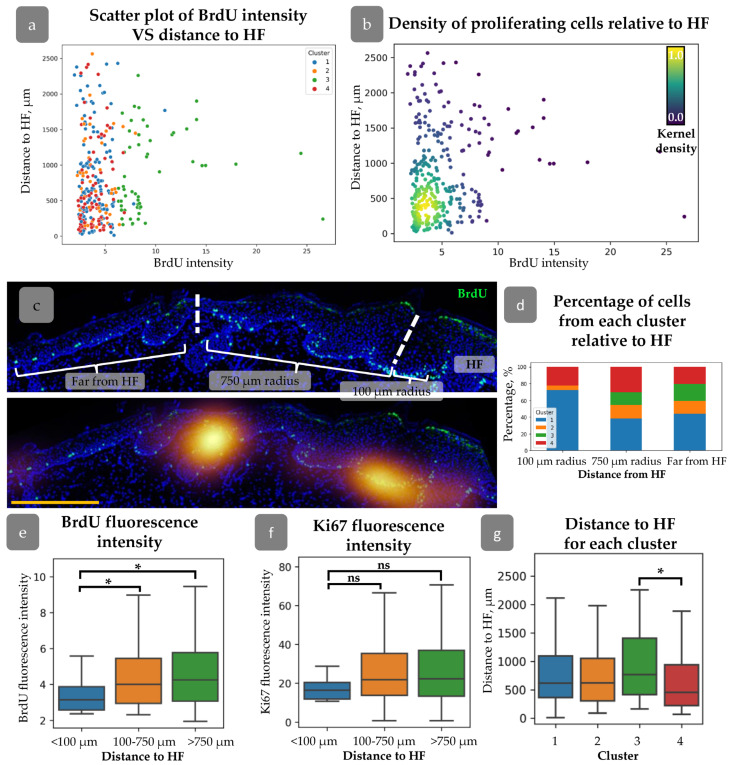
(**a**) Scatter plot of BrdU intensity against the distance to HF for cells from different clusters. (**b**) Density scatter plot of BrdU intensity against the distance to HF (kernel density estimation) showing higher proliferating cell density around HFs. (**c**) Top—an example of cell category identification based on the distance from HF. Bottom—BrdU density map overlaid onto the top image. Dotted lines split the epidermal regions identified based on the distance from HF. Blue staining—DAPI. Scale bar—400 µm. (**d**) Percentage of cells from each cluster at different distances from HF. (**e**,**f**) BrdU (**e**) and Ki67 (**f**) fluorescence intensities for cells located at different distances from HF. (**g**) Mean distance to HF for cells from each cluster. * *p* < 0.05, ns—no significant differences (one-way ANOVA with Tukey correction for multiple comparisons).

**Figure 5 cells-14-00448-f005:**
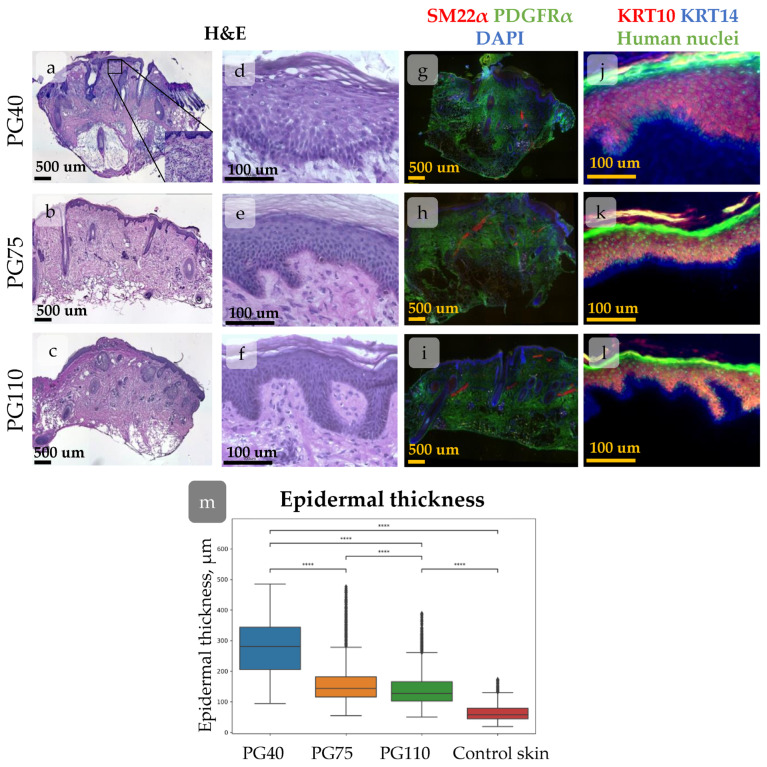
(**a**–**l**) Histological images of xenografted skin at post-grafting days (PG) 40 (**a**,**d**,**g**,**j**), 75 (**b**,**e**,**h**,**k**), and 110 (**c**,**f**,**i**,**l**) showing dynamical morphological changes during regeneration. Scale bar—500 µm or 100 µm. (**a**–**f**) Haematoxylin–eosin staining of xenografted skin. Whole-section images (**a**-**c**) and the enlarged view of the epidermis (**d**–**f**). (**g**–**l**) IHC staining for specific markers. (**g**–**i**) Staining for SM22α (arrector pili muscles), PDGFRα (total dermal fibroblasts). Nuclei are stained with DAPI. (**j**–**k**) Staining for KRT10 (keratinocytes at suprabasal skin layers), KRT14 (keratinocytes at basal skin layer). Nuclei are stained with anti-human nuclei antibodies. (**m**) Epidermal thickness at different timepoints post-xenografting. A trend towards a decrease in mean epidermal thickness during regeneration is observed. ****—*p* < 0.0001 (one-way ANOVA with Tukey correction for multiple comparisons).

**Figure 6 cells-14-00448-f006:**
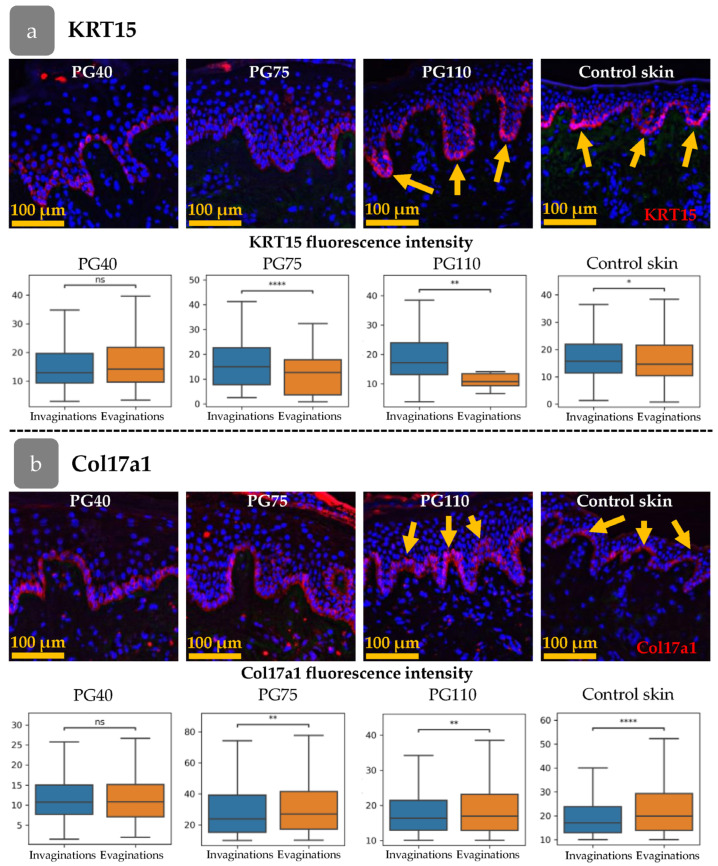
Distribution pattern of KRT15 (**a**) and Col17a1 (**b**) at different stages of xenograft regeneration. Top—IHC staining. The KRT15 or Col17a1 staining (red) overlaid onto DAPI (blue). Scale bar—100 µm. Bottom—fluorescence intensities in epidermal invaginations and evaginations. Yellow arrows indicate the formation of a distinct spatial pattern of marker distribution. *—*p* < 0.05; **—*p* < 0.01; ****—*p* < 0.0001 (Student’s *t*-test).

**Figure 7 cells-14-00448-f007:**
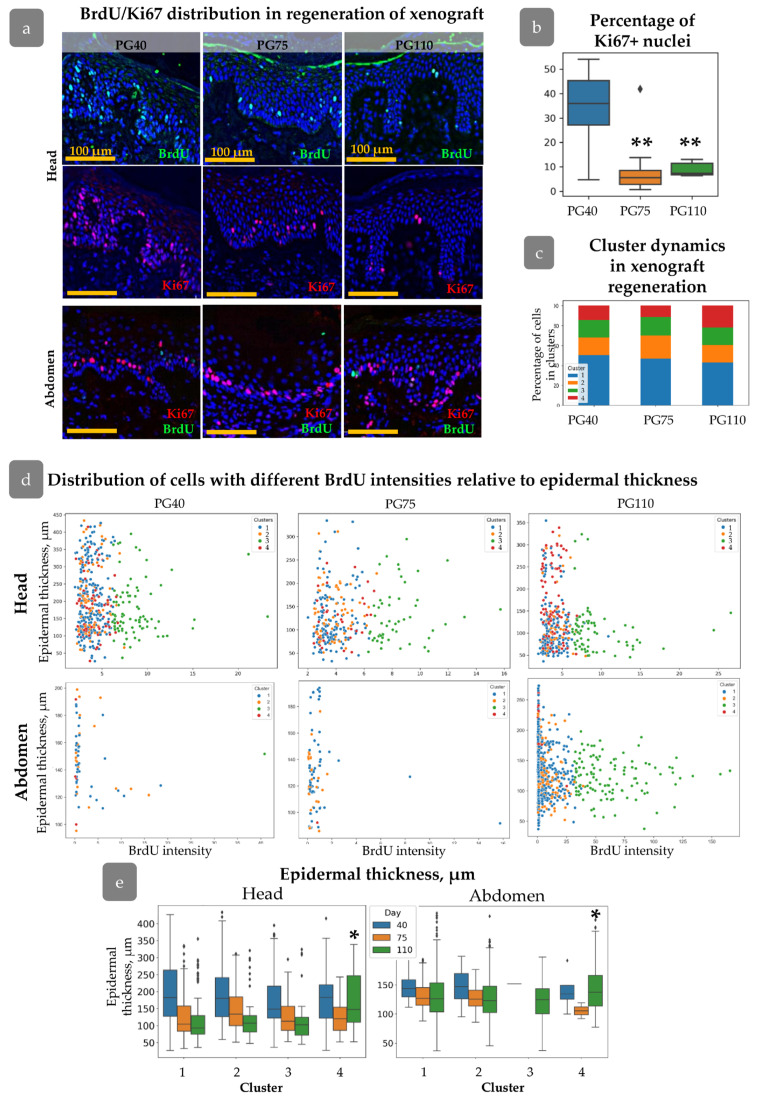
Nuclei parameters during xenograft regeneration. (**a**) IHC staining of skin with BrdU (green), Ki67 (red), and DAPI (blue) at PG 40, 75, and 110 in head and abdominal xenografts. Scale bar: 100 µm. (**b**) Percentage of Ki67+ cells in the basal layer on the PG40-110. Outliers are shown on graph. **—*p* < 0.01 relative to PG40. (**c**) Cluster composition at different timepoints after xenografting. (**d**) Scatter plot of BrdU intensity versus epidermal thickness for cells from different clusters at PG40–110 in head and abdominal xenografts. (**e**) Epidermal thickness for each cluster during regeneration in head and abdominal xenograft, reflecting the acquisition of specific spatial positions by cells related to Cluster 4. Outliers are shown on graph. *—*p* < 0.05. Statistical analysis: ANOVA with Tukey correction for multiple comparisons.

**Figure 8 cells-14-00448-f008:**
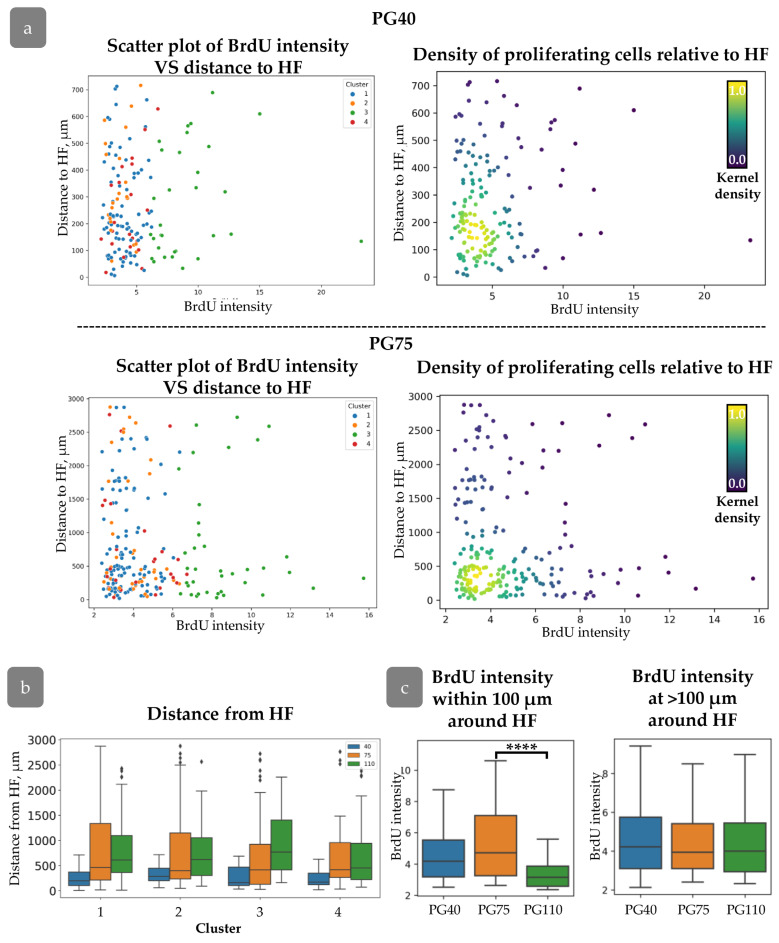
(**a**) Distribution of cells with varying BrdU intensities relative to HFs at PG40 (**top**) and PG75 (**bottom**). (**Left**): scatter plot of BrdU intensity against distance to HF for cells from different clusters. (**Right**): Density scatter plot of BrdU intensity against distance to HF (kernel density estimation). (**b**) Distance to HF for each cluster during regeneration. (**c**) BrdU intensity for cells at different distances from HF. ****—*p* < 0.0001. Statistical analysis: ANOVA with Tukey correction for multiple comparisons.

**Figure 9 cells-14-00448-f009:**
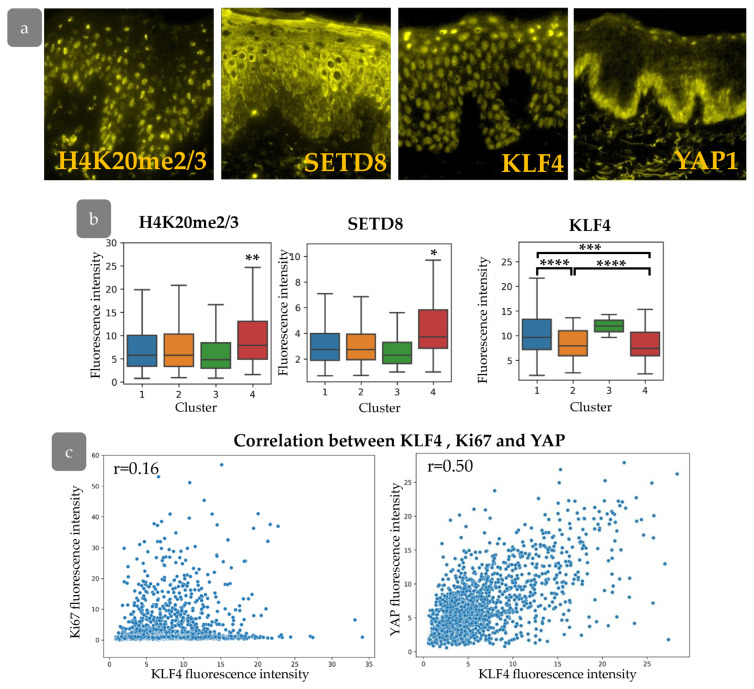
(**a**) IHC staining for H4K20me2/3, SETD8, KLF4, and YAP. (**b**) The expression of H4K20me2/3, SETD8, and KLF4 in epidermal clusters. *—*p* < 0.05; **—*p* < 0.01; ***—*p* < 0.001; ****—*p* < 0.0001 (ANOVA with Tukey correction for multiple comparisons). (**c**) Scatter plot of KLF4 fluorescence intensity against Ki67 (**left**) and YAP (**right**) fluorescence intensities, with the indication of Pearson correlation coefficient (r).

**Figure 10 cells-14-00448-f010:**
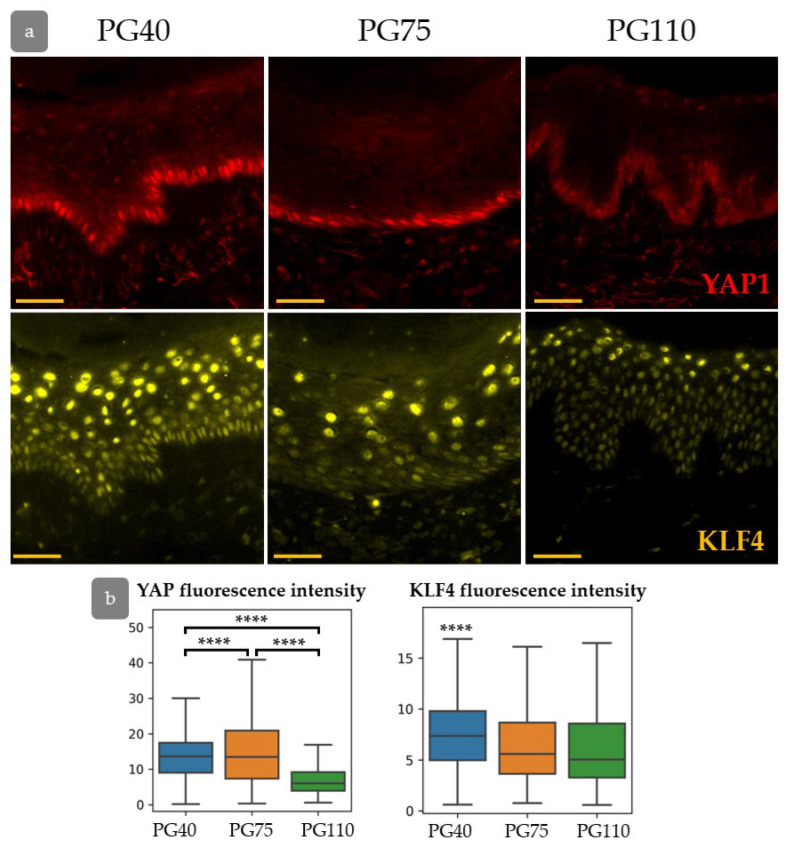
(**a**) IHC staining for YAP (**top**) and KLF4 (**bottom**) in the process of xenograft regeneration. (**b**) The expression of YAP and KLF4. ****—*p* < 0.0001 (ANOVA with Tukey correction for multiple comparisons).

**Figure 11 cells-14-00448-f011:**
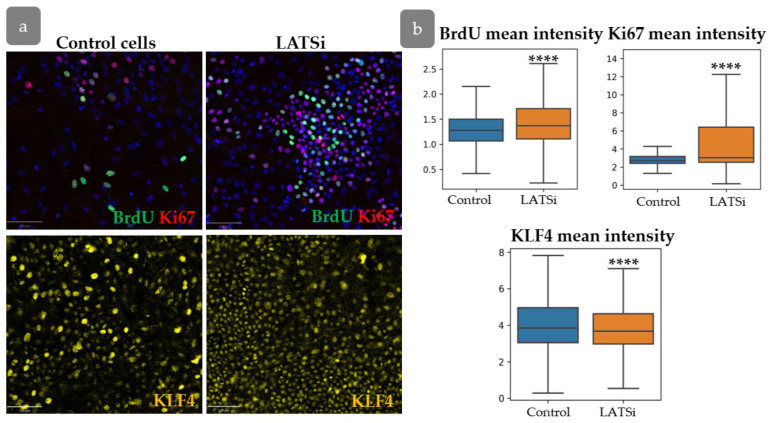
The analysis of primary keratinocyte culture. (**a**) IHC staining of keratinocytes with BrdU, Ki67 (**top**), and KLF4 (**bottom**). Scale bar: 100 µm. (**b**) Measurement of the total BrdU, Ki67, and KLF4 fluorescent intensities in cells in control cells compared to cells with LATSi treatment. ****—*p* < 0.0001 (one-way ANOVA with Tukey correction for multiple comparisons).

**Figure 12 cells-14-00448-f012:**
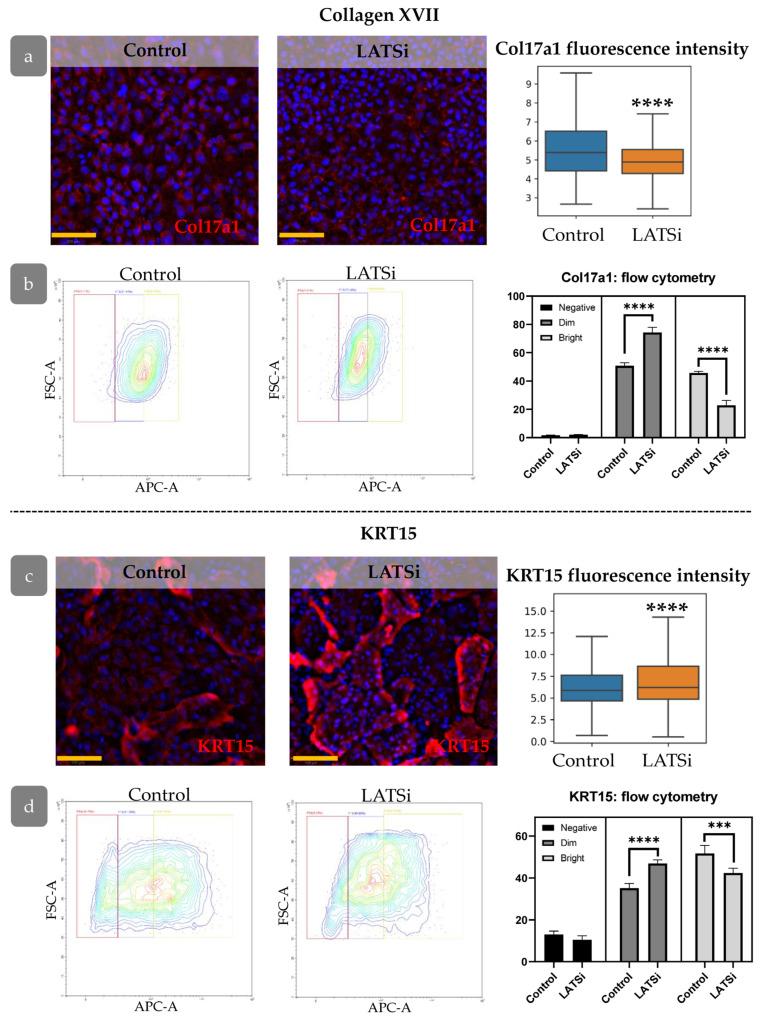
The expression of markers associated with epidermal relief formation in cell culture: (**a**,**b**)—Col17a1, (**c**,**d**)—KRT15, (**a**,**c**) IHC staining for the analysed markers, blue staining—DAPI, scale bar: 100 µm (**left**) and measurement of the overall marker fluorescence intensity in the cell population (**right**). (**b**,**d**) Flow cytometry analysis of primary keratinocytes. (**Left**)—representative flow cytometry contour plot. (**Right**)—the percentages of cells within different categories (negative, dim, and bright). ****—*p* < 0.0001 (one-way ANOVA with Tukey correction for multiple comparisons).

**Figure 13 cells-14-00448-f013:**
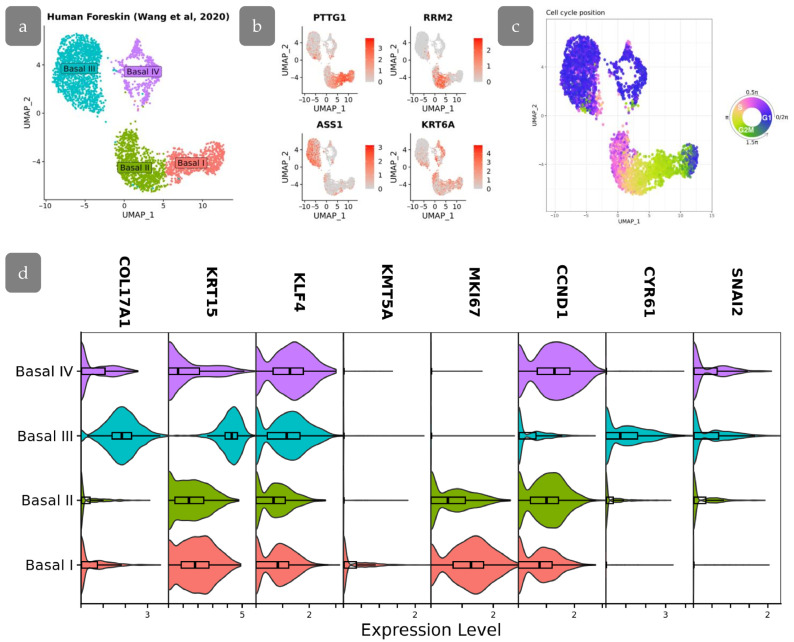
scRNA-seq analysis of human foreskin. (**a**) UMAP projection of basal cell subpopulations based on single-cell RNA-seq data from Wang et al., 2020 [15]. (**b**) Feature plots showing the expression of specific gene markers in basal cell subpopulations. (**c**) Visualization of cell cycle position on UMAP embeddings. (**d**) Stacked violin plots showing the expression of genes of interest among basal cell subpopulations.

**Table 1 cells-14-00448-t001:** Antibodies used for immunohistochemical staining.

Antibody	Cat. №	Dilution
Primary rat anti-BrdU	Abcam, ab6326	1:250
Primary rabbit anti-Ki67	Abcam, ab16667	1:200
Primary rabbit anti-KRT14	Abcam, ab181595	1:1000
Primary mouse anti-KRT10	Thermo Fisher Scientific, MA1-06319	1:200
Primary rabbit anti-KRT15	Abcam, ab52816	1:100
Primary rabbit anti-Col17a1	Novus Biologicals, NBP2-38686	1:100
Primary rabbit anti-PDGFRa	Abcam, ab203491	1:500
Primary goat anti-SM22a	Abcam, ab10135	1:100
Primary mouse anti-histone H4 (di methyl K20, tri methyl K20)	Abcam, ab78517	1:400
Primary mouse anti-KMT5A/SETD8	Abcam, ab3798	1:100
Primary goat anti-KLF4	R&D Systems, Minneapolis, MN, USA, AF3640	1:15
Primary rabbit anti-YAP	Abcam, ab52771	1:100
Secondary goat anti-rat IgG, Alexa 488	Invitrogen, San Diego, CA, USA, A-11006	1:1000
Secondary donkey anti-rabbit IgG, Alexa 546	Invitrogen, A10040	1:1000
Secondary goat anti-rabbit IgG, Alexa 405	Invitrogen, AB_221605	1:1000
Secondary chicken anti-rabbit IgG, Alexa 488	Invitrogen, A-21441	1:1000
Secondary donkey anti-mouse IgG, Alexa 555	Invitrogen, A32773	1:1000
Secondary donkey anti-goat IgG, Alexa 546	Invitrogen, A11056	1:1000
Secondary goat anti-mouse IgG, Alexa 660	Invitrogen, A21055	1:1000
Secondary chicken anti-goat IgG, Alexa 647	Invitrogen, A21469	1:1000
Anti-human nuclei Alexa 488 conjugate	Sigma-Aldrich, MAB1281A4	1:100

**Table 2 cells-14-00448-t002:** Parameters of identified cell clusters.

Cluster	BrdU Intensity	Ki67 Intensity	Nuclei Area	Nuclei Circularity	Localization
1	Low	Low	Medium	Round	Mixed
2	Low	Low	Small	Elongated	Mixed
3	High	High/Low	Medium	Round	Inter-ridges
4	Low	Low/High	Large	Round	Rete ridges

## Data Availability

The raw data supporting the conclusions of this article will be made available by the authors on request.

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
