# Peer review of "The Recovery of Epidermal Proliferation Pattern in Human Skin Xenograft"

_cells, 2025, doi:10.3390/cells14060448_

Round 1
Reviewer 1 Report
Comments and Suggestions for Authors
The manuscript provides an in-depth investigation into epidermal keratinocyte proliferation patterns using a human skin xenograft model. It offers insights into the spatial dynamics of keratinocyte proliferation, the role of rete ridges and hair follicles with regards to their cellular localization. Furthermore, the authors investigate the regenerative capabilities of human skin grafted onto immunodeficient mice. While it aligns with and builds on foundational studies, its use of computational clustering and xenograft models sets it apart. However, incorporating in vivo validation and more molecular insights would enhance its impact further.
The study is well-structured, methodologically rigorous, and addresses an important area of dermatological research. Nevertheless, the reviewer cannot accept the manuscript for publication prior to addressing important points. Please find below the reviewer comments that should be addressed before a publication can be considered:
Major comments:
1) This study leverages human skin xenografts to investigate epidermal proliferation, highlighting the role of rete ridges and hair follicles (HFs) in regulating keratinocyte dynamics. Whereas previous studies (Kalabusheva et al., 2020; Kaufmann et al., 1993) also utilized human skin xenografts, the present study focuses less on structural integration or immune responses in grafts. In contrast, the manuscript builds on those foundations by employing high-resolution imaging and computational analysis for deeper insights into keratinocyte behavior. The reviewer finds that the introduction lacks a strong rationale for this approach. It is advised to provide further arguments for the chosen methodology and better describe how it expands on existing literature. More so, what role does the full restoration of the epidermal niche – as claimed by the authors – play when compared to models that lack these stem cells?
2) Moreover, recent work by Cherkashina et al. (2023) introduced a similar model but did not classify proliferative cells into distinct clusters as comprehensively as done here. How does the present work build on and significantly extend that previous work?
3) Shen et al. (2023) recently discussed the morphogenesis and functional roles of rete ridges, noting their potential as stem cell niches. This study's use of xenografts strengthens the argument for rete ridges’ involvement in epidermal homeostasis. However, this is done on a purely phenotypical level, using indirect readouts from correlation studies. The authors should support their findings with a description of the underlying mechanistics.
4) The cluster analysis (Figure 6) should be extended by incorporating an identification of the various identifiable cell populations. This could be done by spatial RNASeq, IHC or related techniques. It is crucial to know if cells originate from the same or different cell populations. And in case of the latter, what subpopulations could be identified. The reviewer is not convinced whether the differences in the proliferative profile are random or are based on spatial differences or cell-cell interactions (homo- or heterotypic), paracrine signaling or other effects. A similarly likely explanation could be proximity to blood vessels or afferent nerves. A more in-depth investigation that sheds some light on the underlying mechanisms is crucial. The distance from hair follicles or epidermal thickness do not seem to be significant factors and the reviewer is not convinced that the presented evidence goes beyond mere correlation.
5) Similarly, the evidence shown in Figure 7c does not show a strong spatial bias for BrdU staining.
a. Note 1: legend is missing what colors are indicating (red, green; blue is apparently Nuclear counterstain, red presumably is BrdU)
b. Note 2: the authors should explain what the large black box signifies. What data is covered?
6) Rather than fluorescence intensity, it would be interesting to see how many Brdu+ or Ki67+ cells are located within the different distances away from the HF.
7) While morphological and proliferation patterns are well-studied, molecular pathways regulating these dynamics (e.g., signaling cascades) are not explored in as much detail as in studies like Mascré et al. (2012). The study misses out on delving into the molecular pathways or signaling mechanisms driving the observed proliferation patterns. The authors should therefore incorporate a more in-depth analysis of the cellular dynamics for the cell populations under investigation. In particular, the exploration of pathways (e.g., Wnt, Notch, or Hedgehog signaling) that regulate keratinocyte behavior could strengthen the biological context and relevance.
8) The reviewer is missing a longitudinal molecular analysis during the described stages of regeneration. Although the temporal aspects of xenograft regeneration are well-described, molecular profiling (e.g., transcriptomic or proteomic changes) during different regeneration stages is missing, which could provide deeper insights into the processes involved.
9) Validation of findings in additional models (e.g., organoid cultures or direct human skin biopsies) would enhance robustness.
10) The manuscript does not critically address the limitations of using xenografts, such as the absence of physiological immune responses or differences in graft-host interactions compared to native human skin.
11) The potential effects of mouse-specific factors (e.g., dermal environment) on the proliferation dynamics observed in the xenografts are not discussed.
12) While the study briefly mentions pathological conditions like psoriasis and impaired wound healing, it does not explicitly link its findings to potential therapeutic applications.
13) Practical suggestions for leveraging the insights (e.g., improving skin graft techniques, treating hyperproliferative disorders) are not thoroughly explored.
14) The rationale for selecting specific parameters (e.g., BrdU/Ki67 intensity, nuclear morphology) for clustering keratinocytes is not sufficiently justified. It is unclear how these parameters align with the study’s hypotheses or biological significance.
15) The exclusion of other potential factors (e.g., spatial proximity to rete ridges or HFs) from clustering raises questions about the completeness of the analysis and should be addressed in more detail.
16) Overall, figure legends lack sufficient detail for standalone comprehension.
17) The study does not investigate how aging or specific skin conditions (e.g., chronic wounds, dermatitis) might alter proliferation dynamics in xenografts, limiting its translational relevance.
18) Although statistical methods are described, effect sizes or additional metrics (e.g., confidence intervals) are not consistently reported, which could help contextualize the significance of findings.
19) The study does not address the reproducibility of clustering results or validate them against alternative methods.
20) The discussion does not fully integrate findings with the broader body of literature on keratinocyte biology, rete ridges, or HF-associated stem cells.
21) Contradictory models (e.g., single-progenitor versus hierarchical) are mentioned but not critically evaluated in light of the study's findings.
22) In general, the results section puts too much emphasis on methodological descriptions. These parts should be transferred to the methods part and a stronger emphasis should be put on a description focused on results instead.
Author Response
The authors are grateful for the valuable comments provided by the reviewer. To strengthen the presented results, we have expanded our investigations and included additional details regarding the molecular mechanisms underlying epidermal proliferation. Specifically, we have elucidated the role of YAP signaling as a key driver of epidermal regeneration and confirmed its influence using a cellular model. Furthermore, we have performed a bioinformatical analysis of a published single-cell dataset to more precisely characterize basal epidermal populations. We believe that these improvements have enhanced the reliability of the manuscript and provide a more in-depth understanding of the subject matter.
Comments 1: This study leverages human skin xenografts to investigate epidermal proliferation, highlighting the role of rete ridges and hair follicles (HFs) in regulating keratinocyte dynamics. Whereas previous studies (Kalabusheva et al., 2020; Kaufmann et al., 1993) also utilized human skin xenografts, the present study focuses less on structural integration or immune responses in grafts. In contrast, the manuscript builds on those foundations by employing high-resolution imaging and computational analysis for deeper insights into keratinocyte behavior. The reviewer finds that the introduction lacks a strong rationale for this approach. It is advised to provide further arguments for the chosen methodology and better describe how it expands on existing literature. More so, what role does the full restoration of the epidermal niche – as claimed by the authors – play when compared to models that lack these stem cells?
Response 1: We sincerely appreciate your thoughtful comments and suggestions. Whilst the preceding studies sought to characterise the morphology of transplanted skin and address the question of whether the model could represent normal conditions, the objective of the present study was to examine the proliferation pattern in human epidermis. To this end, skin xenografts were employed as a model that reproduces relatively normal epidermal morphology. A description of the applications of this model was included in the Introduction section (lines 79-84):
The grafting of human skin into immunodeficient mice has shown the restoration of most skin structures, confirmed the comparative biomechanical and physiological proper-ties of xenograft and therefore provides a functional in vivo model for the investigation of human skin [15–17]. The xenograft has been served as a representative model for studying pathologies [18–23], treatment options [24–27] and the processes occurring in normal state [28].
The potential limitations of the xenograft model were also discussed in the Discussion section (lines 927-939):
The utilisation of human skin xenografts facilitated the identification of stem and progenitor cell populations within the epidermis. While many studies consider the xeno-graft model to be the normal one, some aspects of the model should be taken into account when interpreting the results. These include the infiltration of human graft with mouse cells [74], the participation of the host cells in regeneration processes [75] as well as the neovascularization which occurs with the help of mouse endothelial cells [76], and the loss of human resident immune cells [77]. The present study has demonstrated that the skin xenograft model exhibited activated YAP signalling even at PG110, which may be the un-derlying cause of the thickened epidermis observed in comparison to normal skin. These findings suggest that the partial inhibition of YAP activity in future xenograft studies could potentially lead to the restoration of a normal phenotype. The utilisation of different skin sources enabled the identification of analogous cell populations, thereby indicating the persistence of cell types even within a system that excluded mouse cells.
It is proposed that skin xenografts may serve as a more reliable model than mouse models or cellular 2D or 3D models, which lack the human stem cell niche.
Comments 2: Moreover, recent work by Cherkashina et al. (2023) introduced a similar model but did not classify proliferative cells into distinct clusters as comprehensively as done here. How does the present work build on and significantly extend that previous work?
Response 2: We are grateful for your careful consideration of our previous work. The aim of our previous work (Cherkashina et al., 2022) was to study the regeneration of human skin after xenotransplantation. In that work we described the process of xenograft regeneration with the precise attention to the morphological changes and marker distribution in the regenerating skin. Based on that work we were able to conclude that xenografted human skin becomes morphologically closer to the normal state at post-grafting day 110 and therefore used this term in our further analysis as the model of comparatively normally functioning skin.
In our current work, we focused on the study of epidermal proliferation patterns. While human models of skin regeneration are restricted, we have chosen the xenograft model as the one that highly represents the actual tissue functioning. Our current work aims to get an insight into epidermal proliferation mechanisms in human skin which are not covered by the previous article.
Comments 3: Shen et al. (2023) recently discussed the morphogenesis and functional roles of rete ridges, noting their potential as stem cell niches. This study's use of xenografts strengthens the argument for rete ridges’ involvement in epidermal homeostasis. However, this is done on a purely phenotypical level, using indirect readouts from correlation studies. The authors should support their findings with a description of the underlying mechanistics.
Response 3: Thank you for highlighting key areas for improvement. We have extended our study by describing the expression of specific markers that can indicate the proliferative status of epidermal keratinocytes and are associated with stemness and exit from the stem cell niche. These markers include di-/tri-methylated H4K20, SETD8, KLF4. The expression of di-/tri-methylated H4K20 and SETD8 was associated with the population of progenitors located in the rete ridges, while KLF4 allowed the identification of stem cell clusters uniformly distributed in the epidermal basal layer. We have also shown the importance of YAP signalling modulation in the formation of spatially distinct cell populations. These data (described in chapters 3.9 “Cluster definition”, 3.11 “Transcriptional differences in proliferating cell populations”) should describe the mechanistics of cell proliferation hierarchy more precisely.
Comments 4: The cluster analysis (Figure 6) should be extended by incorporating an identification of the various identifiable cell populations. This could be done by spatial RNASeq, IHC or related techniques. It is crucial to know if cells originate from the same or different cell populations. And in case of the latter, what subpopulations could be identified. The reviewer is not convinced whether the differences in the proliferative profile are random or are based on spatial differences or cell-cell interactions (homo- or heterotypic), paracrine signaling or other effects. A similarly likely explanation could be proximity to blood vessels or afferent nerves. A more in-depth investigation that sheds some light on the underlying mechanisms is crucial. The distance from hair follicles or epidermal thickness do not seem to be significant factors and the reviewer is not convinced that the presented evidence goes beyond mere correlation.
Response 4: We thank the reviewer for the insightful feedback. We recognize the importance of investigating the identity and microenvironment of the cell clusters identified in our analysis and have taken several steps to address these points.
As suggested, we have extended our cluster analysis by incorporating the expression of markers associated with stemness and differentiation, namely, di-/tri-methylated H4K20, SETD8, and KLF4. This analysis, detailed in [Chapter 3.9 “Cluster definition”], has allowed us to further characterize the proliferative status of the observed cell clusters and gain insights into their potential lineage relationships. To further validate the identity of our clusters, we performed an analysis of the published single-cell RNA sequencing dataset from Wang et al. (2020). This analysis revealed parallels between the identified clusters in our xenografted skin and transcriptionally defined clusters of basal epidermal cells, providing further support for the relevance and accuracy of our clustering. These findings are detailed in [Chapter 3.11 “Transcriptional differences in proliferating cell populations”].
To assess the potential influence of hair follicles on proliferation patterns, as suggested by the reviewer, we compared the proliferation of cells in head skin xenografts (containing hair follicles) with that in abdominal skin xenografts (which lack hair follicles). The observed significant differences in proliferation patterns between these skin sources confirm the role of hair follicles in modulating epidermal proliferative activity. This comparison is presented in [Chapter 3.5. “Clustering of Proliferative Cells in Epidermal Basal Layer” ].
We acknowledge the widely discussed question regarding the role of rete ridges in epidermal renewal and the potential influence of underlying structures such as vasculature. Directly determining whether epidermal relief serves as an epidermal niche, or is simply a consequence of differing proliferation activity, is beyond the scope of this study. The association of vasculature with epidermal proliferation pattern is a quite complicated question as its association with the epidermal relief is also well-documented (Lawlor et al., 2015). As such, we concentrated on the dynamics of rete ridge formation and the spatial positioning of proliferating cells during the regeneration process, providing valuable insights into the morphological aspects of epidermal renewal in our model. We intend to address the complexities of rete ridge mechanistics, including the interplay with vasculature, paracrine signaling or other effects in future research, as this requires a separate, focused investigation.
Comments 5: Similarly, the evidence shown in Figure 7c does not show a strong spatial bias for BrdU staining.
a. Note 1: legend is missing what colors are indicating (red, green; blue is apparently Nuclear counterstain, red presumably is BrdU)
b. Note 2: the authors should explain what the large black box signifies. What data is covered?
Response 5: Thank for your careful check of the provided images. We changed the figure that more clearly reflected the obtained result. In order to prove spatial BrdU distribution aspects we constructed a density map reflecting BrdU+ cell density distribution in tissue. The changes are in line 468, Figure 7c
5a. We added the legend explaining colors in the image.
5b. In order to obtain images of full tissue slices, we created panoramic images. However, in order to reduce image size and avoid significant delays in analysis capacity due to large memory requirements, we performed panoramic imaging only on the epidermis. Consequently, the majority of the dermis remained outside the field of view. The large black box corresponded to the panoramic borders, covering no significant data. However, we modified the image to provide more information about BrdU distribution, and the revised image does not have this issue.
Comments 6: Rather than fluorescence intensity, it would be interesting to see how many Brdu+ or Ki67+ cells are located within the different distances away from the HF.
Response 6: We appreciate the reviewer's suggestion to analyze the number of BrdU+ or Ki67+ cells at different distances from the hair follicle (HF). To address the reviewer's request, we would like to direct their attention to Figure 7a,b, line 468. Here, the density of points provides a visual representation of cell quantity at specific distances from the HF. We have carefully considered the value of displaying the data as the scatter plot versus binning the data into histograms. We have decided to leave the original graph as is, due to the loss of details and the fact that the existing plot represents the most accurate and non-biased representation of our data.
Comments 7: While morphological and proliferation patterns are well-studied, molecular pathways regulating these dynamics (e.g., signaling cascades) are not explored in as much detail as in studies like Mascré et al. (2012). The study misses out on delving into the molecular pathways or signaling mechanisms driving the observed proliferation patterns. The authors should therefore incorporate a more in-depth analysis of the cellular dynamics for the cell populations under investigation. In particular, the exploration of pathways (e.g., Wnt, Notch, or Hedgehog signaling) that regulate keratinocyte behavior could strengthen the biological context and relevance.
Response 7: We are grateful for incredibly helpful comments. We have extended our analysis with the study of YAP signaling activity and its interplay with KLF4 expression during the regeneration skin xenograft. To expand our understanding of the role of YAP in the regulation of proliferation patterns we studied the effect of YAP activation in the culture of primary keratinocytes. Our results show that activation of YAP cascade led to the activated proliferation with more active BrdU label incorporation. The analysis was supplemented by the single cell data analysis which allowed us to propose that YAP hyperactivation prevents the specific positioning of proliferating cells. This data is described in Chapters 3.9 “Cluster definition”, 3.10 “YAP influence on cell proliferation dynamics”, 3.11 “Transcriptional differences in proliferating cell populations”.
Comments 8: The reviewer is missing a longitudinal molecular analysis during the described stages of regeneration. Although the temporal aspects of xenograft regeneration are well-described, molecular profiling (e.g., transcriptomic or proteomic changes) during different regeneration stages is missing, which could provide deeper insights into the processes involved.
Response 8: We sincerely appreciate your suggestions and extended the analysis. We traced the expression of YAP and KLF4 in the regenerating human skin xenograft which allowed us to explore their role in the formation of epidermal relief and proliferation pattern. These results are described in Chapter 3.9 “Cluster definition”. We also confirmed YAP participation in proliferation pattern establishment on cellular model (Chapter 3.10 “YAP influence on cell proliferation dynamics”) and the formation of epidermal relief using single cell data analysis (Chapter 3.11 “Transcriptional differences in proliferating cell populations”).
Comments 9: Validation of findings in additional models (e.g., organoid cultures or direct human skin biopsies) would enhance robustness.
Response 9: Authors appreciate the recommendation. We decided to validate our findings by studying the proliferation activity of primary human keratinocytes in normal state and with activation of YAP signaling pathway. We describe the results obtained on primary keratinocyte culture in Chapter 3.10 “YAP influence on cell proliferation dynamics”.
Comments 10: The manuscript does not critically address the limitations of using xenografts, such as the absence of physiological immune responses or differences in graft-host interactions compared to native human skin.
Response 10: We are grateful for the valuable suggestion. We added the discussion of potential limitations of the xenograft model which could be found in lines 927-939:
The utilisation of human skin xenografts facilitated the identification of stem and progenitor cell populations within the epidermis. While many studies consider the xenograft model to be the normal one, some aspects of the model should be taken into account when interpreting the results. These include the infiltration of human graft with mouse cells [73], the participation of the host cells in regeneration processes [74] as well as the neovascularization which occurs with the help of mouse endothelial cells [75], and the loss of human resident immune cells [76]. The present study has demonstrated that the skin xenograft model exhibited activated YAP signalling even at PG110, which may be the underlying cause of the thickened epidermis observed in comparison to normal skin. These findings suggest that the partial inhibition of YAP activity in future xenograft studies could potentially lead to the restoration of a normal phenotype. The utilisation of different skin sources enabled the identification of analogous cell populations, thereby indicating the persistence of cell types even within a system that excluded mouse cells.
Comments 11: The potential effects of mouse-specific factors (e.g., dermal environment) on the proliferation dynamics observed in the xenografts are not discussed.
Response 11: We are grateful for the suggestion. The Discussion section was expanded to include a discussion of the potential mouse-specific effects that occur in xenograft models. The implications of these effects on the study's findings is also considered (lines 927-939).
Comments 12: While the study briefly mentions pathological conditions like psoriasis and impaired wound healing, it does not explicitly link its findings to potential therapeutic applications.
Response 12: We are grateful for the suggestion. We emphasized in our with on the understanding of normal skin function and then discussed the possibility of the findings practical application in the Discussion section (lines 891-905)
A number of dermatological conditions are linked to abnormalities in cell prolifera-tion kinetics. An understanding of normal physiological processes is paramount to eluci-dating the pathogenesis of disease. In this context, our results highlighting the critical role of YAP in maintaining epidermal homeostasis are particularly significant. The precise reg-ulation of YAP activity is essential for proper epidermal function, and disruptions to this regulation are increasingly implicated in various skin pathologies. The hyperactivation of YAP is associated with psoriasis [63,64], tumors such as squamous cell carcinoma [65–67] and basal cell carcinoma [68–70] while YAP deficiency is often associated with epider-molysis bullosa [71], or may prevent chronic wounds from healing. in conditions such as epidermolysis, diabetes, and ageing. [72]. Therefore, the insights gained from our study, demonstrating contribution of YAP to epidermal homeostasis under normal conditions, provide a crucial framework for interpreting the alterations in YAP activity observed in diseased states. The identification of the characteristics that underpin epidermal homeo-stasis enables the formulation of strategies for reestablishing the skin's normal state in pathological conditions.
Comments 13: Practical suggestions for leveraging the insights (e.g., improving skin graft techniques, treating hyperproliferative disorders) are not thoroughly explored.
Response 13: We are grateful for the comment and added the suggestions concerning YAP pathway modulation for the improvement of xenograft model to the Discussion section (lines 933-939):
The present study has demonstrated that the skin xenograft model exhibited activated YAP signalling even at PG110, which may be the underlying cause of the thickened epi-dermis observed in comparison to normal skin. These findings suggest that the partial in-hibition of YAP activity in future xenograft studies could potentially lead to the restora-tion of a normal phenotype. The utilisation of different skin sources enabled the identifica-tion of analogous cell populations, thereby indicating the persistence of cell types even within a system that excluded mouse cells.
Comments 14: The rationale for selecting specific parameters (e.g., BrdU/Ki67 intensity, nuclear morphology) for clustering keratinocytes is not sufficiently justified. It is unclear how these parameters align with the study’s hypotheses or biological significance.
Response 14: We are grateful for the mindful examination of the described techniques. The analysis was initiated with a discussion of the measured variables and their potential to function as clustering parameters in chapters 3.3. “Parameters for Cell Clustering: BrdU Intensity” and 3.4. “Parameters for Cell Clustering: Nuclear Shape”. While BrdU and Ki67 staining reflect the proliferation activity of the cells, we based on the staining of skin xenograft cryosections with both these markers. We also utilized nuclear shape as a parameter for clusterisation as it has been shown that nuclear shape reflects cell proliferation and differentiation state. In addition, nuclear shape was utilised as a parameter for clusterisation, as it has been demonstrated that nuclear shape reflects cell proliferation and differentiation state. A concise overview of this analytical stage is provided in the Introduction (lines 90-93):
We selected and analysed specific nuclear features, tested their potential to characterise stemness and proliferation capacity of epidermal keratinocytes and then selected the clus-tering parameters for machine learning algorithms to analyse skin sections stained for BrdU and Ki67.
and the Discussion with emphasis placed on the parameter selection process (lines 699-703):
The application of a designed algorithm based on QuPath software enabled the de-scription of proliferative pattern in human epidermis. Initially, the parameters that could be extracted from histological images were analysed. Skin sections were stained for BrdU and Ki67 which mark the proliferating cells. It was also demonstrated that nuclear shape can be used as a parameter reflecting the proliferative state of the cell (Figure 5).
In order to confirm the results of nuclei clustering, we further studied the distribution of di-/tri-methylated H4K20, SETD8, KLF4 in the identified cell clusters (Chapter 3.9 “Cluster definition”). This analysis allowed us to confirm the biological significance of our findings.
Comments 15: The exclusion of other potential factors (e.g., spatial proximity to rete ridges or HFs) from clustering raises questions about the completeness of the analysis and should be addressed in more detail.
Response 15: The primary objective of this study was to identify distinct subpopulations of cells based on their intrinsic morphological characteristics, independent of their location within the epidermis. Incorporating spatial proximity directly into the k-means clustering algorithm would have risked artificially grouping cells based solely on their location, rather than their inherent cellular features. This could have masked or confounded the identification of truly distinct subpopulations defined by morphology. To achieve a more comprehensive and unbiased understanding of the cellular heterogeneity within the epidermis, it was necessary to exclude spatial proximity from the clustering algorithm and subsequently analyse the spatial distribution of the identified clusters.
We expanded the description of such an approach in lines 384-394:
Epidermal thickness and distance to the HF were not included in the clustering parameters to avoid artificially clustering of populations based on their position in the epidermis, but further these parameters were counted for each cell cluster to verify whether distinct pop-ulations with particular morphological characteristics can occupy specific locations. In-corporating spatial proximity directly into the k-means clustering algorithm would have risked artificially grouping cells based solely on their location, rather than their inherent cellular features.
Comments 16: Overall, figure legends lack sufficient detail for standalone comprehension.
Response 16: We appreciate the reviewer’s feedback regarding the figure legends. In response, we have expanded the descriptions for figure legends to provide more comprehensive information. If further details or clarifications are needed, we are willing to provide additional information as required.
Comments 17: The study does not investigate how aging or specific skin conditions (e.g., chronic wounds, dermatitis) might alter proliferation dynamics in xenografts, limiting its translational relevance.
Response 17: We thank the reviewer for raising the important point regarding the potential influence of aging and skin conditions on the proliferation dynamics in our xenograft model. We acknowledge that these factors can significantly impact epidermal behavior. To address this, we would like to clarify the characteristics of the donor skin used in our study: all donor skin samples were obtained from patients undergoing elective plastic surgery procedures. This ensured that the tissue was free from pre-existing skin pathologies such as dermatitis, chronic wounds, or other conditions known to alter epidermal function. We utilized tissue from several different donors. Importantly, our analysis revealed only minimal differences in epidermal proliferation patterns between the different donors, suggesting a degree of consistency across the tissue samples used. While we observed minimal differences between donors, we did identify model-specific differences in epidermal proliferation. This suggests that the observed variations are primarily driven by the model-specific conditions (e.g. skin source), rather than inherent donor variability.
Comments 18: Although statistical methods are described, effect sizes or additional metrics (e.g., confidence intervals) are not consistently reported, which could help contextualize the significance of findings.
Response 18: We appreciate your suggestion to deepen the statistical analysis. The groups we compared were identified through cluster analysis and calculating effect sizes may not accurately reflect the differences between these complex groupings, as they do not represent a straightforward treatment effect. In the context of cluster analysis, effect sizes can be challenging to interpret meaningfully. The clusters may vary significantly in their characteristics, and a single effect size metric may not capture the multifaceted nature of these differences. Instead, we have focused on descriptive statistics and visual representations (box plots, scatter plots) which provide a clearer understanding of the relationships among clusters.
Comments 19: The study does not address the reproducibility of clustering results or validate them against alternative methods.
Response 19: The reproducibility of clustering results was confirmed by the comparison of clustering results of separate samplings within the full dataset. We also compared cell clusters identified in head and abdominal xenograft. All the data has shown similarities with the only differences reflecting differences of proliferation patterns in different models. We added the description of cluster composition in abdominal skin in chapter 3.5. “Clustering of Proliferative Cells in Epidermal Basal Layer” (lines 427-434):
Clusters found in abdominal xenografts slightly differed from those in head region, however possessed some common morphological characteristics. The epidermal relief in the abdominal skin was less pronounced compared to head skin, however the differences between clusters were observed. Clusters 1 and 2 were similar to that observed in head xenograft. Cluster 3 contained rarely-dividing nuclei with high BrdU but low Ki67 inten-sity. Epidermal thickness for the Cluster 3 was low indicating its localization in in-ter-ridges. By the way, high Ki67 staining was identified for cells in Cluster 4 which occu-pied predominantly rete ridges (Figure 6d).
We also added single cell data analysis which showed the existence of the similar transcriptionally distinct clusters in human skin. The analysis is described in Chapter 3.11 “Transcriptional differences in proliferating cell populations”.
Comments 20: The discussion does not fully integrate findings with the broader body of literature on keratinocyte biology, rete ridges, or HF-associated stem cells.
Response 20: We thank the reviewer for the comment regarding the integration of our findings with the broader literature. To address this, we have significantly expanded the discussion section to better contextualize our results within the existing body of knowledge.
We have provided a more detailed description of the alterations observed in specific markers characterizing the stem status of cells, such as KLF4, SETD8, di-/tri-methylated H4K20, YAP (Chapters 3.9 “Cluster definition”, 3.10 “YAP influence on cell proliferation dynamics”, 3.11 “Transcriptional differences in proliferating cell populations”). This expanded analysis allows us to better understand the proliferation and differentiation patterns in the context of our experimental model. We have strengthened the connection between our findings and the existing literature on the role of rete ridges and HFs in maintaining epidermal homeostasis. We have provided a more detailed discussion of how our results align with previous studies highlighting the importance of these structures as stem cell niches and their influence on epidermal regeneration. We also discussed the potential of the observed pattern to correspond to the hypothesis of a single progenitor which is described in lines 940-952.
The obtained data allow us to propose that the single progenitor hypothesis [5] is largely consistent with our results. The single cluster corresponding to the less differenti-ated cells, which could be referred to as stem cells, is distributed uniformly in the epider-mis.The clusters that were identified as being associated with the epidermal relief possess the features of the more differentiated cells, thus matching the described differentiation trajectories from the single stem cluster to spatially distinct populations [33]. The hypoth-esis of two distinct stem cell populations with different proliferation rates [8] is rendered less probable due to the differentiation status of the spatially-associated clusters. Further-more, this finding aligns with another study performed on a xenograft model, which ob-served no spatial proximity of proliferating clones [28]. Consequently, the present findings suggest the necessity for further investigation into the mechanisms that govern the for-mation of proliferation hierarchies, given that a single stem-like population gives rise to spatially segregated populations exhibiting differential proliferation rates.
By incorporating these changes, we believe that the revised discussion section provides a more comprehensive and nuanced interpretation of our results, strengthening the overall impact and relevance of our study.
Comments 21: Contradictory models (e.g., single-progenitor versus hierarchical) are mentioned but not critically evaluated in light of the study's findings.
Response 21: We are grateful for the suggestion and added the discussion of the contradictory models to the Discussion section (lines 940-952):
The obtained data allow us to propose that the single progenitor hypothesis [5] is largely consistent with our results. The single cluster corresponding to the less differenti-ated cells, which could be referred to as stem cells, is distributed uniformly in the epider-mis.The clusters that were identified as being associated with the epidermal relief possess the features of the more differentiated cells, thus matching the described differentiation trajectories from the single stem cluster to spatially distinct populations [33]. The hypoth-esis of two distinct stem cell populations with different proliferation rates [8] is rendered less probable due to the differentiation status of the spatially-associated clusters. Further-more, this finding aligns with another study performed on a xenograft model, which ob-served no spatial proximity of proliferating clones [28]. Consequently, the present findings suggest the necessity for further investigation into the mechanisms that govern the for-mation of proliferation hierarchies, given that a single stem-like population gives rise to spatially segregated populations exhibiting differential proliferation rates.
Comments 22: In general, the results section puts too much emphasis on methodological descriptions. These parts should be transferred to the methods part and a stronger emphasis should be put on a description focused on results instead.
Response 22: We appreciate the reviewer's comment regarding the balance between methodological descriptions and results presentation in the Results section. While we understand the concern about potential redundancy, we believe that, in this particular case, the inclusion of certain methodological aspects within the Results section is essential for a comprehensive and nuanced interpretation of our findings. The techniques employed in this study are complex and not widely standardized. Providing a detailed description of these aspects alongside the presentation of the results allows the reader to fully understand the nuances of how the data was generated and analyzed, which is crucial for assessing the validity and reliability of our conclusions. Without a clear understanding of the methodological steps, there is a risk that the reader may misinterpret the results or draw incorrect conclusions. By providing a more comprehensive description, we aim to minimize this risk and ensure that the reader can fully grasp the significance of our findings.
Reviewer 2 Report
Comments and Suggestions for Authors
In this manuscript, the authors investigate skin cells with different proliferation states by analyzing nuclear morphology, BrdU/Ki67 staining patterns, and incorporating computational image analysis. While the manuscript presents a substantial amount of data, the presentation appears somewhat disorganized, and the conclusions drawn from the data are not clearly articulated. The reviewers’ specific comments are as follows:
1. Immunofluorescent staining in Fig 2e-2h are blurry, please replace them with higher-resolution images.
2. The staining images in Figure 1 and Figure 3a were from the same sample and same region. Please replace them with staining images from different regions.
3. Please label the BrdU+ Ki67+ cells, BrdU+ Ki67− cells, and BrdU− Ki67+ cells in Figure 3a.
4. In the absence of Ki-67 staining, what criteria did the authors use to identify and label the four different clusters in Figure 6d? Additionally, please provide a clearer version of Figure 6d to facilitate readers' understanding and interpretation.
5. How are the expression levels of the basal keratinocyte marker Krt14 and the committed progenitor marker Involucrin distributed across these four clusters? In the conclusion section, the authors state that two clusters represent stem cells, while the other two represent progenitor cells. What evidences support this classification?
6. Please label the protein names represented by the different fluorescent colors in the figure 7c. Additionally, the bottom right corner of this figure is obscured by a large black square.
7. Page 8, line 254: Authors cannot directly copy sentences from the cited paper. A better approach is to rephrase the original content in your own words while citing the source.
8. Page 4 line 154 has a typo in this sentence “distance do the annotations with basal membrane”.
9. Page 8 line 235-245, references are not found.
10. Page 9 line 270,273, references are not found.
Author Response
We are grateful to the reviewer for the valuable comments. In response, we have strengthened our results by expanding our investigations and adding details about the molecular mechanisms of epidermal proliferation. Our revised manuscript now highlights YAP signaling as a key aspect driving epidermal regeneration, confirmed through cellular modeling. Additionally, we conducted a bioinformatical analysis of a published dataset to further investigate basal epidermal populations. We believe these improvements have made the manuscript more reliable and provided a more in-depth analysis of the subject.
Comments 1. Immunofluorescent staining in Fig 2e-2h are blurry, please replace them with higher-resolution images.
Response 1: Thank you for noticing the image issues, we changed an images that are more clear
Comments 2. The staining images in Figure 1 and Figure 3a were from the same sample and same region. Please replace them with staining images from different regions.
Response 2: We are thankful for your careful check. We changed the image 3a with the image from different region (line 287)
Comments 3. Please label the BrdU+ Ki67+ cells, BrdU+ Ki67− cells, and BrdU− Ki67+ cells in Figure 3a.
Response 3: Thank you for the suggestion. We marked these cell types on the image (line 287)
Comments 4. In the absence of Ki-67 staining, what criteria did the authors use to identify and label the four different clusters in Figure 6d? Additionally, please provide a clearer version of Figure 6d to facilitate readers' understanding and interpretation.
Response 4: We are grateful for your careful consideration of the performed analysis. A unique number was assigned to each nucleus detected in QuPath. Following cluster analysis, the object numbers were employed to identify the nuclei in the initial image. Consequently, the clusters depicted in figure 6d represent those detected by cluster analysis. We added this explanation in the text for better understanding (lines 406-408):
A unique number was assigned to each nucleus detected in QuPath. Following cluster analysis, the object numbers were employed to identify the nuclei in the initial image. Consequently, we were able to depict the clusters in epidermis (Figure 6f).
On the previous version of the figure only BrdU staining was depicted, however the original image contained information for Ki67 staining as well. We provided the clearer version with the BrdU and Ki67 staining. After re-structuring the text this figure got number 6f (line 439).
Comments 5. How are the expression levels of the basal keratinocyte marker Krt14 and the committed progenitor marker Involucrin distributed across these four clusters? In the conclusion section, the authors state that two clusters represent stem cells, while the other two represent progenitor cells. What evidences support this classification?
Response 5: We sincerely thank the reviewer for the question. In normal human skin Krt14 is distributed along the basal layer while Involucrin is observed only starting from the granular epidermal layer. Instead we decided to trace the distribution of histone H4 methylation at lysine 20 along with the methyltransferase SETD8 associated with H4K20 monomethylation. We also studied KLF4 distribution in the identified clusters. This gave us information about the lowest differentiation level of cells from Cluster 2 and indicated the onset of terminal differentiation for cells from Cluster 4. The results are described more precisely at chapter 3.9 “Cluster definition”.
Comments 6. Please label the protein names represented by the different fluorescent colors in the figure 7c. Additionally, the bottom right corner of this figure is obscured by a large black square.
Response 6: We are grateful for the suggestion. We changed the figure that more clearly reflected the obtained result. In order to prove spatial BrdU distribution aspects we constructed a density map reflecting BrdU+ cell density distribution in tissue. We also added the legend explaining colors in the image.
In order to obtain images of full tissue slices, we created panoramic images. However, in order to reduce image size and avoid significant delays in analysis capacity due to large memory requirements, we performed panoramic imaging only on the epidermis. Consequently, the majority of the dermis remained outside the field of view. The large black square corresponded to the panoramic borders, covering no significant data. However, we modified the image to provide more information about BrdU distribution, and the revised image does not have this issue.
Comments 7. Page 8, line 254: Authors cannot directly copy sentences from the cited paper. A better approach is to rephrase the original content in your own words while citing the source.
Response 7: Authors apologize for the carelessness in describing the observations from the sources used. We have corrected the phrase in the following way (lines 339-342):
The alterations in nuclear morphology that occur following the onset of keratinocyte differentiation have been previously described: keratinocytes located in the basal epider-mal layer exhibit vertically oriented ellipsoid nuclei and acquire a horizontally oriented spheroid form during the process of terminal differentiation [22,23].
Comments 8. Page 4 line 154 has a typo in this sentence “distance do the annotations with basal membrane”.
Response 8: We have corrected the typo. You can find the sentence in the line 201:
“distance to the annotations with basal membrane”
Comments 9. Page 8 line 235-245, references are not found.
Comments 10. Page 9 line 270,273, references are not found.
Response 9, 10: We have corrected some technical issues in the document and hope that the references now are displayed correctly.
Round 2
Reviewer 1 Report
Comments and Suggestions for Authors
Overall, the authors have partially managed to incorporate the reviewer's suggestions. They have expanded their introduction, provided additional experimental validation, clarified their methodologies, improved figure legends, and acknowledged limitations. However, a few areas still need better justification or additional analysis.
Below is a detailed breakdown of each comment and response:
Comment 1: The authors have clarified the model's utility, but they have not explicitly compared how their methodology expands on previous literature in a way that sets it apart significantly. The reviewer suggests that a more explicit discussion is included that addresses how their computational analysis and high-resolution imaging uniquely advance prior findings would strengthen the argument.
Comment 6: The authors’ justification for not binning data into histograms is reasonable, but they did not explicitly show how different cell types distribute relative to HF distance. Thus, a density plot or additional statistical test confirming non-random distribution would further strengthen the claim.
Comment 8: While YAP/KLF4 tracking is considered useful, broader transcriptomic or proteomic changes are still missing. Hence, even if a full transcriptomic analysis is not feasible, the authors could discuss temporal changes in proliferation markers would enhance the response.
Comment 17: While the variability was minimized, the effects of aging or disease conditions remain unexplored. The reviewer suggests that a brief outlook is included that discusses how these factors could be tested in future work.
Comment 22: While the reviewer agrees that methodological details are important, some sections still read more like methods than results. Therefore, the authors shouldshift some some detailed explanations to (Supplementary) Materials.
Author Response
We appreciate your thoughtful feedback and the opportunity to strengthen our submission. The manuscript has been revised in accordance with the suggestions provided. The chapters containing the technical details have been relocated to the Appendix section, several graphs have been incorporated, and the Discussion section has been expanded. All the changes are highlighted in the manuscript with yellow. We have also prepared detailed responses to each of your comments, which are included below.
Comment 1: The authors have clarified the model's utility, but they have not explicitly compared how their methodology expands on previous literature in a way that sets it apart significantly. The reviewer suggests that a more explicit discussion is included that addresses how their computational analysis and high-resolution imaging uniquely advance prior findings would strengthen the argument.
Response 1: We appreciate the reviewer's suggestion to further clarify how our methodology expands on previous literature. To address this, we have extended the introduction with the following addition (lines 79-81):
The in vivo tracing of human epidermal cells would provide valuable insights into proliferation patterns. This is particularly relevant given ongoing debates about stem cell dynamics, such as the existence of spatially distinct stem cell populations with different proliferation rates [8] or the hypothesis of a single progenitor [33]. While vital tracers have enabled the identification of proliferating cells and informed conclusions about their dynamics, and mathematical modeling based on computational analysis of H2B-traced epidermal cells has described proliferation patterns in the epidermis [33], these studies have primarily relied on mouse models combined with human biopsy analysis. However, these models do not allow for real-time proliferation tracing in human skin, and biopsies only capture a static state. This question leads to the necessity of the development of in vivo tracing approaches in human tissues. In combination with computational analysis of the obtained high-resolution images, it may extend the understanding of epidermal regeneration mechanisms.
Additionally, we have enhanced the discussion by including the following paragraph (lines 989-993):
The combination of in vivo labelling followed by computational analysis with high-resolution imaging allowed the identification of several clusters of proliferating cells in the epidermis. The precise hypothesis regarding epidermal stem cell organisation remains uncertain; it encompasses the hypothesis of two stem cell populations [8] and the hypothesis of a single progenitor [15], which both have gained support. The obtained data allow us to propose that the single progenitor hypothesis [5] is largely consistent with our results….
Comment 6: The authors’ justification for not binning data into histograms is reasonable, but they did not explicitly show how different cell types distribute relative to HF distance. Thus, a density plot or additional statistical test confirming non-random distribution would further strengthen the claim.
Response 6: We appreciate the reviewer's suggestion to provide additional visual and statistical evidence regarding the distribution of different cell types relative to hair follicle (HF) distance. To address this, we have enhanced our figures with new graphs.
Specifically, Figure 4g (line 445) illustrates the tendency of cells from Cluster 3 to localize further away from HFs, while cells from Cluster 4 are concentrated near HFs.
Furthermore, we have added new graphs to Figure 8 (line 573). Figure 8b demonstrates that the correlation between proliferation activity and distance from HF is consistent across all clusters, with an increase in "Distance to HF" corresponding to an increase in xenograft size. Additionally, Figure 8c shows an increase in proliferation activity within a 100 µm radius of HF, as indicated by a decrease in mean BrdU intensity. These additions provide further evidence supporting our claims and strengthen the argument regarding the distribution and behavior of cell types in relation to HF distance.
Comment 8: While YAP/KLF4 tracking is considered useful, broader transcriptomic or proteomic changes are still missing. Hence, even if a full transcriptomic analysis is not feasible, the authors could discuss temporal changes in proliferation markers would enhance the response.
Response 8: We are grateful for the suggestion. We have addressed the reviewer's comment by extending the discussion to include a more detailed explanation of the skin regeneration process and the role of YAP/KLF4 activity within that process (lines 897-908).
In the initial phases of skin regeneration, the role of YAP is considered to be of significance in the process of skin restoration [57,58] supporting active proliferation [59], migration [60,61] and preventing apoptosis [62]. However, it has been demonstrated that persistent activity of YAP results in the induction of epithelial-mesenchymal transition [63,64], an increase in proinflammatory cytokine and growth factor levels [65–68], and the onset of tumors [45]. Our data indicate the important role of YAP in the control of proliferation and formation of the epidermal relief and associated spatially distinct cell clusters. The activity of KLF4 is also important for the proper regeneration process [69]. While YAP has been shown to negatively regulate KLF4 expression [41], the overexpression of KLF4 may induce YAP activation in regeneration [70]. The gradual decline in KLF4 expression during xenograft regeneration, concomitant with the decrease in YAP activity (Figure 13), is a prerequisite for the normal epidermal differentiation.
Comment 17: While the variability was minimized, the effects of aging or disease conditions remain unexplored. The reviewer suggests that a brief outlook is included that discusses how these factors could be tested in future work.
Response 17: To address the Reviewer’s suggestion, we have added a paragraph to the discussion that outlines potential future directions for exploring these factors. Specifically, we have included the following text (lines 981-988):
It is also worthy of note that the regenerative potential of xenografted skin is significantly influenced by factors such as donor age and individual particularities, even in cases involving the use of skin from healthy donors undergoing cosmetic surgeries. To ensure the attainment of reproducible results, it is imperative to select donors of similar age and without underlying pathologies. The investigation of age-related disparities in skin regeneration following xenotransplantation and the assessment of the influence of pathological conditions on this process merit further exploration in future studies.
Additionally, we have extended the description of YAP role in epidermal regeneration by highlighting the potential of xenografts to serve as models for pathological conditions. This enhancement provides a clearer path for future studies to investigate how aging and disease might influence the processes we have described (lines 950-954).
A number of dermatological conditions are linked to abnormalities in cell proliferation kinetics. An understanding of normal physiological processes is paramount to elucidating the pathogenesis of disease. In this context, our results highlighting the critical role of YAP in maintaining epidermal homeostasis are particularly significant. The precise regulation of YAP activity is essential for proper epidermal function, and disruptions to this regulation are increasingly implicated in various skin pathologies. The hyperactivation of YAP is associated with psoriasis [77,78], tumors such as squamous cell carcinoma [64,79,80] and basal cell carcinoma [81–83] while YAP deficiency is often associated with epidermolysis bullosa [84], or may prevent chronic wounds from healing in conditions such as epidermolysis, diabetes, and ageing [57]. Therefore, the insights gained from our study, demonstrating contribution of YAP to epidermal homeostasis under normal conditions, provide a crucial framework for interpreting the alterations in YAP activity observed in diseased states. The identification of the characteristics that underpin epidermal homeostasis enables the formulation of strategies for reestablishing the skin's normal state in pathological conditions. Furthermore, xenotransplantation of human skin opens up broad prospects for disease modelling [16]. A range of models have been developed, including those of acute wounds [85–87], pressure ulcers [21,88], psoriasis [89,90] and viral infections [20,22]. These models are intended to facilitate the subsequent study of treatment options identified in the model of normal regeneration.
Comment 22: While the reviewer agrees that methodological details are important, some sections still read more like methods than results. Therefore, the authors should shift some detailed explanations to (Supplementary) Materials.
Response 22: We appreciate the reviewer's persistence in highlighting this issue and are grateful for the opportunity to revisit our approach. Upon further consideration, we agree with the reviewer's suggestion and have shifted chapters 3.2-3.4 to the Appendix section to ensure that the main text focuses more on results rather than detailed methodological explanations.
Reviewer 2 Report
Comments and Suggestions for Authors
The authors have addressed all my questions.
Author Response
We appreciate your review of our manuscript. As part of our revision process, we have made several improvements to enhance the clarity and impact of our work. Specifically, we have relocated chapters containing technical details to the Appendix section, incorporated several new graphs, and expanded the Discussion section. All the changes are highlighted in the manuscript with yellow. Thank you for your time and consideration.